# FedMAP: Unlocking Potential in Personalized Federated Learning through Bi-Level MAP Optimization

## Abstract

Federated Learning (FL) enables collaborative training of machine learning (ML) models on decentralized data while preserving data privacy. However, data across clients often differs significantly due to class imbalance, feature distribution skew, sample size imbalance, and other phenomena. Using information from these not identically distributed (non-IID) datasets causes challenges in training. Existing FL methods based on a single global model cannot effectively capture client data variations, resulting in suboptimal performance. Personalized FL (PFL) techniques were introduced to adapt to the local data distribution of each client and utilize the data from other clients. They have shown promising results in addressing these challenges. We propose FedMAP, a novel Bayesian PFL framework which applies Maximum A Posteriori (MAP) estimation to effectively mitigate various non-IID data issues, by means of a parametric prior distribution, which is updated during aggregation. We provide a theoretical foundation illustrating FedMAP's convergence properties. In particular, we prove that the prior updates in FedMAP correspond to gradient descent iterations for a linear combination of envelope functions associated with the local losses. This differs from previous FL approaches, that aim at minimizing a weighted average of local loss functions and often face challenges with heterogeneous data distributions, resulting in reduced client performance and slower convergence in non-IID settings. Finally, we show, through evaluations of synthetic and real-world datasets, that FedMAP achieves better performance than the existing methods. Moreover, we offer a robust, ready-to-use framework to facilitate practical deployment and further research.

## 1 Introduction

By leveraging distributed data sources, ML models can be trained more effectively and produce robust, generalized insights. Nevertheless, stringent privacy restrictions, security threats, and high data transfer costs have made centralized methods impossible, particularly in sensitive sectors such as healthcare (Zhang et al., 2024). FL was introduced as a practical paradigm that enables collaborative training across clients without exchanging raw data.

In practice, data across organizations are often non-IID, which is one of the main challenges to successfully adopting FL. Non-IID data refers to data that is not identically distributed across clients, resulting in challenges, including slower model convergence, reduced model accuracy, and increased communication costs in FL. Li et al. (2022) identified three main categories of non-IID data: label distribution skew, feature distribution skew, and quantity skew. In real-world settings, clients often experience combinations of different types of non-IID data. For instance, a rural clinic might have fewer patient records (quantity skew) and collect data using different medical equipment (feature distribution skew), and serve a population with different disease prevalence rates (label distribution skew) compared to urban hospitals. Addressing the combination of these non-IID factors is necessary for the practical and effective deployment of FL systems.

Classic FL algorithms such as FedAvg (McMahan et al., 2016), rely on Maximum Likelihood Estimation (MLE) principles and assume that local client updates can be aggregated to obtain a single global model that maximizes the likelihood of all clients' data collectively. This assumption fails

with non-IID data, which the global objective function is unable to accurately represent individual client's data distribution (Li et al., 2021b). This discrepancy creates a fundamental optimization challenge in which local gradients from different clients can point in different directions extensively and result in slow convergence or poor local optima. Although the variants of FedAvg have made progress in handling data heterogeneity, they usually struggle to capture the complexities of non-IID data. PFL was introduced to learn personalized models customized to each client's unique data distribution and leverage the collective knowledge across clients (Kulkarni et al., 2020). However, existing PFL methods face several issues, which include inefficient knowledge transfer between clients, high communication costs, and limited personalization capabilities (Lin et al., 2022). These limitations are all due to the underlying optimization approaches that fail to capture the complexities of non-IID data distributions.

We introduce FedMAP, a new PFL framework that fundamentally addresses the FL challenges of non-IID data by utilizing a global prior distribution derived from MAP estimation. We formulate the FL problem as a bi-level optimization that adaptively learns and updates a global prior distribution that guides local model optimization. This approach can effectively balance knowledge sharing and local data distribution adaptation. FedMAP tackles the core optimization issues of client drift and the inconsistency between global and local objectives in non-IID settings. Our contributions include a mathematical framework for the PFL using MAP estimation, the FedMAP algorithm with an adaptive weighting scheme for robust global prior updates which specifically mitigate the practical scenarios where clients experience a combination of non-IID data distributions, empirical evaluation on both synthetic and public datasets that incorporate various non-IID data distributions, and the integration of FedMAP with the open-source Flower FL framework (Beutel et al., 2020). The evaluation results show FedMAP consistently outperforms individual client training and the existing FL methods. Furthermore, the theoretical analysis indicates that FedMAP converges to the solution of a bi-level optimization problem, providing a foundation for its effectiveness in handling non-IID data distributions. With its improved reproducibility, FedMAP is available for community use and further development. The code is available at https://anonymous.4open.science/r/FedMAP-963/.

## 1.1 Related literature

**Approaches for addressing non-IID issues through standard FL** One of the key challenges in classic FL is the statistical heterogeneity in clients' data. This often leads to large variations in model performance (Vahidian et al., 2024). Several solutions have been proposed to mitigate the impact of this heterogeneity. FedProx (Li et al., 2020) adds a proximal term to the local optimization process, limiting how far local updates can deviate from the global model. SCAFFOLD (Karimireddy et al., 2020) utilizes control variates to correct client drift in local updates. Despite these improvements, relying on a single global model schema often fails to generalize across the diverse data distributions of different clients, leading to suboptimal performance in many cases (Kulkarni et al., 2020).

**Personalized federated learning** Recent research highlights the PFL can eliminate the impact of data heterogeneity by customizing the global model to individual clients' data distributions(Kulkarni et al., 2020). The personalization can be achieved in several ways. *Fine-Tuning* (Marfoq et al., 2022; Lee et al., 2023) allows clients to adjust a global model locally using their own data. *Layer-wise Personalization* involves personalizing specific layers of the network such as the batch normalization layers while sharing other layers (Li et al., 2021b). FedASA (Deng et al., 2024) was proposed to use an adaptive cell-wise architecture selection strategy to determine which layers to share based on client heterogeneity. PerAda (Xie et al., 2024) introduces a parameter-efficient approach using adapters for personalization while maintaining generalization through knowledge distillation. *Multi-Task Learning* treats each client as a unique task, optimizing across related tasks to improve overall model performance (Smith et al., 2017). pFedEM (Chen et al., 2024) extends this by modelling each client's data distribution as a time-varying mixture of multiple base distributions. *Meta-Learning* strategies such as Model-Agnostic Meta-Learning (MAML) (Fallah et al., 2020) construct a model to adapt to new client data with minimal retraining. *Cluster-Based Personalization* (Ren et al., 2023; Porcu et al., 2022) groups clients by data similarity, each cluster developing a shared model based on its common characteristics. These strategies collectively balance robust global model learning with effective local adaptation, thus optimizing FL across diverse environments. *Regularization-based Personalization* adds regularization terms to the learning objective to control the deviation between the local models and a global model (Li et al., 2021a).

**Bayesian approach in FL** Adopting Bayesian methods in FL offers several benefits. Particularly these methods can effectively handle non-IID data by quantifying uncertainty, enhancing robustness through evidence-based likelihood estimates, and improving performance on limited data using prior distributions for each model parameter (Cao et al., 2023). The method pFedBayes (Zhang et al., 2022) introduces weight uncertainty in neural networks by balancing the loss on private data with the divergence from a global variational distribution using variational Bayesian inference. $\beta$-Predictive Bayes (Hasan et al., 2024) improves calibration and uncertainty estimation by approximating the global predictive posterior through interpolating a mixture and a product of local predictive posteriors using a tunable parameter $\beta$. FedPop (Kotelevskii et al., 2022) conceptualizes FL as population modelling, using Markov Chain Monte Carlo methods for federated stochastic optimization and accounting for data heterogeneity with common population parameters and random effects. Furthermore, the Bayesian nonparametric framework proposed in (Yurochkin et al., 2019) models local neural network weights and applies a Beta-Bernoulli process-based inference technique to synthesize a global network from local models without additional data pooling.

Building upon the challenges and solutions discussed above, FedMAP combines Bayesian principles with regularization-based PFL. FedMAP shares similarities with Ditto (Li et al., 2021a) on the methods to balance local specialization with global knowledge sharing. However, Ditto solves a bi-level optimization problem with separate objectives for global and local models. FedMAP uses MAP estimation for local optimization, directly integrating global knowledge into the local objective. pFedMe (T Dinh et al., 2020) is another similar approach to FedMAP, which uses Moreau envelopes as regularizers in bi-level optimization. With Gaussian prior, FedMAP reduces to pFedMe's formulation, however FedMAP is more generic with its flexible choice of prior distributions. Also, FedMAP differs from pFedBayes which uses variational inference. FedMAP uses MAP estimation since it is more computationally efficient when incorporating prior knowledge. Similar to FedPop's population modelling concept, FedMAP allows each client to have a personalized model and benefit from the collective knowledge encoded in the global prior. Nevertheless, FedMAP uses a more straightforward probabilistic framework with an adaptive weighting mechanism in the global aggregation step, which considers the confidence and relevance of each client's model instead of sample size. Most of the mentioned approaches only address isolated non-IID issues, whereas FedMAP is designed to address the combination of non-IID data distributions which are common in practical scenarios.

## 2 FROM MLE TO MAP ESTIMATOR

Let us consider a prediction task in which the input space is denoted by $\mathcal{X}$ and the output space is $\mathcal{Y}$. As mentioned in the introduction, in the FL framework, one has available not only one but a collection of datasets

$$Z_k = \{(x_k^{(i)}, y_k^{(i)})\}_{i=1}^{N_k} \in (\mathcal{X} \times \mathcal{Y})^{N_k}, \qquad \text{for} \quad k = 1, \ldots, q,$$

each of them consisting of an IID sampling from a probability distribution $\mathcal{D}_k$.

The learning procedure in the FL framework consists of two stages, which can be repeated iteratively:

1. *Local training:* Each client trains a model based on its local data and the global model (if a global model is available).

2. *Aggregation:* The central server constructs (or updates) a global model based on the outcome of the local training at each local client.

Each iteration of this two-stage process is known as a *communication round*. Both the local training and the aggregation can be seen as particular examples of learning tasks. Since we are looking for a probabilistic model $\phi : \mathcal{X} \to \mathcal{P}(\mathcal{Y})$, a typical choice for the loss functional during the local training is the negative log-likelihood, defined as

$$\mathcal{L}(\phi, (x, y)) := -\log \mathbb{P}(y|\phi(x)),$$

where $\mathbb{P}(y|\phi(x))$ denotes the likelihood of the random variable $y$ associated to the probability distribution $\phi(x)$. Let us consider a parameterized family of models

$$\mathcal{H} := \{\phi(\cdot; \theta) : \mathcal{X} \to \mathcal{P}(\mathcal{Y}) \, : \, \theta \in \Theta\}, \quad \text{where } \Theta \subset \mathbb{R}^d \text{ is the parameter space.}$$

Considering empirical risk minimization as the learning algorithm, each local model can be written as $\phi_k^* = \phi(\cdot\,;\theta_k^*)$, where $\theta^* \in \Theta$ the solution to the minimization problem

$$\theta_k^* \in \arg\min_{\theta \in \Theta} \frac{1}{N_k} \sum_{i=1}^{N_k} -\log\left[\mathbb{P}\left(y_k^{(i)}|\phi(x_k^{(i)};\theta)\right)\right]. \tag{1}$$

This corresponds to a maximum likelihood estimator MLE within the hypothesis set $\mathcal{H}$. The problem of (1) is that the trained model $\phi_k^*$ only depends on the local data, and no knowledge is leveraged across the datasets, which is precisely the goal in FL. For this reason, in most FL methods, one does not aim at solving (1). Instead, one would consider the minimization problem (1), using the global model as an initial guess but never letting the algorithm reach the minimum. The issue of forgetting the global model during the local training can be overcome by using the global model, not only as the initialization of the minimization algorithm but also in the loss function during the local training.

## 2.1 LOCAL TRAINING AS MAP ESTIMATION

In our FL approach, we formulate the local training problem as a MAP estimation of the local models, in which the global model acts as a prior distribution on the class of functions $\mathcal{H}$. Let us consider a parameterized family of probability measures over the parameter space $\Theta$, denoted by

$$\mathcal{G} := \{\rho_\gamma \in \mathcal{P}(\Theta) : \quad \gamma \in \Gamma\},$$

where $\Gamma \subset \mathbb{R}^p$ is the parameter space for the global model.

Given a global model $\rho_\gamma \in \mathcal{G}$, we can compute, for any $(x,y) \in \mathcal{X} \times \mathcal{Y}$ and $\phi(\cdot;\theta) \in \mathcal{H}$, the likelihood of the posterior probability distribution with respect to the prior $\rho_\gamma$. Up to a multiplicative constant, which is independent of $\theta$, we can use Bayes Theorem to write the posterior as

$$\mathbb{P}(\theta|(x,y)) \propto \mathbb{P}((x,y)|\theta)\rho_\gamma(\theta) = \mathbb{P}(y|\phi(x;\theta))\rho_\gamma(\theta).$$

Then, if we consider the problem of minimizing the negative log-likelihood of the posterior probability distribution, we can take the following loss function:

$$\mathcal{L}(\theta, (x,y), \rho_\gamma) := -\log\left[\mathbb{P}(y|\phi(x;\theta))\right] - \log\rho_\gamma(\theta).$$

Denoting by $\gamma^{(t)}$ the parameter of the global model after the $(t-1)$-th communication round, the parameter $\theta_k^{(t)}$ for the $k$-local model, based on $\gamma^{(t)}$, can be obtained as

$$\theta_k^{(t)} \in \arg\min_{\theta \in \Theta} \Big\{ \underbrace{\sum_{i=1}^{N_k} -\log\left[\mathbb{P}(y_k^{(i)}|\phi(x_k^{(i)};\theta))\right]}_{\mathcal{L}(\theta; Z_k)} \underbrace{-\log\rho_{\gamma^{(t)}}(\theta)}_{\mathcal{R}(\theta, \gamma^{(t)})} \Big\}, \tag{2}$$

where we recall that $Z_k = \{(x_k^{(i)}, y_k^{(i)})\}_{i=1}^{N_k} \in (\mathcal{X} \times Y)^{N_k}$ is the dataset of the $k$-th client. The term $\mathcal{R}(\theta, \gamma^{(t)})$ in (2) can be seen as a parametric regularization term.

## 2.2 ESTIMATING THE PRIOR DURING AGGREGATION

Given the local models $(\theta_1, \ldots, \theta_q) \in \Theta^q$, the parameter $\gamma \in \Gamma$ for the global prior can be obtained by minimizing the function

$$\gamma \mapsto \sum_{k=1}^q w_k \mathcal{R}(\theta_k, \gamma) = -\sum_{k=1}^q w_k \log\rho_\gamma(\theta_k),$$

for some weights $(w_1, \ldots, w_q) \in (0,1)^q$ such that $\sum_k w_k = 1$. Using the interpretation of $\rho_\gamma$ as a parametric prior, this can be seen as a weighted sum of the negative likelihoods of $\gamma$ given the local models $\theta_k$ with $k = 1, \ldots, q$. At the $t$-th communication round, given the current global parameter $\gamma^{(t)}$ and the local models $\theta_k^{(t)}$ obtained as in (2), the parameter $\gamma^{(t)}$ is updated by applying gradient descent to the above function, i.e.

$$\gamma^{(t+1)} = \gamma^{(t)} - \lambda \sum_{k=1}^q w_k \nabla_\gamma \mathcal{R}(\theta_k^{(t)}, \gamma^{(t)}), \tag{3}$$

where $\lambda > 0$ is the learning rate. We observe that, during the local training in (2), the local data only appears in the term $\mathcal{L}(\theta; Z_k)$, and therefore, no local data is transmitted to the central node during the aggregation step in (3).

## 2.3 FL AS BI-LEVEL OPTIMIZATION

Next, we address the asymptotic analysis of the iterations given by (2)–(3). More precisely, we prove in Theorem 1 that the updates of the global parameter $\gamma^{(t)}$, obtained by iterating (2) and (3), correspond to gradient descent iterations for the function

$$M(\gamma) := \sum_{k=1}^{q} w_k M_k(\gamma; Z_k), \quad \text{where} \quad M_k(\gamma; Z_k) = \min_{\theta \in \Theta} \left\{ \mathcal{L}(\theta; Z_k) + \mathcal{R}(\theta, \gamma) \right\}. \quad (4)$$

The function $M_k(\gamma; Z_k)$ can be seen as an envelope function associated with the function $\mathcal{L}(\theta; Z_k)$ and the regularizer $\mathcal{R}(\theta, \gamma)$. In the special case of a quadratic regularizer of the form $\mathcal{R}(\theta, \gamma) = \|\theta - \gamma\|^2$, the function $M_k(\gamma; Z_k)$ is the Moreau envelope of $\mathcal{L}(\cdot; Z_k)$. Minimizing a linear combination of Moreau envelopes in the FL setting was proposed in T Dinh et al. (2020). However, we stress that our proposed approach is much more general. See section A.2 for further details.

Under suitable convexity assumptions on $\mathcal{L}(\theta; Z_k)$ and $\mathcal{R}(\theta, \gamma)$, we also prove in Theorem 1 that the function $M(\gamma)$ is strongly convex, and therefore, one can ensure that $\gamma^{(t)}$ given by alternating (2) and (3) converges to the unique minimizer of $M(\gamma)$. Moreover, this minimizer is the solution to the bi-level optimization problem

$$\underset{\theta_k \in \Theta}{\text{minimize}} \, \mathcal{L}(\theta_k; Z_k) + \mathcal{R}(\theta_k, \gamma^*) \quad \forall k = 1, \ldots, q \quad \text{s.t.} \quad \gamma^* \in \arg\min_{\gamma \in \Gamma} \left( \sum_{k=1}^{q} w_k \mathcal{R}(\theta_k, \gamma) \right). \quad (5)$$

Therefore, local training (2) and aggregation (3) can be seen as an alternating strategy to approximate the solution to the above bi-level optimization problem. Local training would address the upper-level problems, whereas the aggregation step addresses the lower-level problems. Incorporating the global model in the loss function for the local training couples the upper- and lower-level optimization problems. This coupling leverages the knowledge of the local models across the clients.

**Theorem 1.** *Let $\Theta \subset \mathbb{R}^d$ be compact and $\Gamma = \mathbb{R}^d$. For each $k = 1, \ldots, q$, let $\mathcal{L}(\theta; Z_k)$ be continuous and convex w.r.t. $\theta$, and let $\mathcal{R}(\theta, \gamma)$ be differentiable and strictly convex in $\Theta \times \Gamma$. Then, the iterations (2)–(3) can be written as*

$$\gamma^{(t+1)} = \gamma^{(t)} - \lambda \nabla_\gamma M(\gamma^{(t)}),$$

*where $M(\gamma)$ is given by (4). Moreover, $M(\gamma)$ is strictly convex in $\Gamma$, and its unique minimizer is the solution to the bi-level optimization problem (5).*

The proof of this result is given in Appendix A. Minimizing a linear combination of envelope functions such as $M(\gamma)$ to train a global model differs from most FL approaches, which focus on minimizing a linear combination of the local loss functions, i.e. $F(\theta) = \sum_{k=1}^{q} w_k \mathcal{L}(\theta; Z_k)$. As we show in section A.4, through a simple example, minimizing $M(\gamma)$ and minimizing $F(\theta)$ may produce completely different global models, especially in the case of non-IID data.

## 3 PROPOSED *FedMAP* ALGORITHM

We propose FedMAP (Federated Maximum A Posteriori), a novel FL algorithm incorporating a global prior distribution over the local model parameters, enabling personalized FL. In the sequel, we will consider, as a hypothesis set for the local models, a family of NNs denoted by

$$\phi(\cdot; \theta): \mathcal{X} \to \mathcal{Y}, \qquad \theta \in \Theta := \mathbb{R}^d.$$

As hypothesis set for the global model, we will consider a Gaussian prior on the parameter space, where the parameter $\gamma$ represents the mean of the distribution. For a fixed parameter $\sigma^2 > 0$, we consider the parameterized family of probability distributions with density function

$$\theta \in \Theta \longmapsto \rho_\gamma(\theta) = \frac{1}{\sqrt{2\pi\sigma^2}} e^{-\frac{\|\theta - \gamma\|^2}{2\sigma^2}}, \qquad \text{for } \gamma \in \Gamma = \Theta. \quad (6)$$

This choice for the parametric prior yields a quadratic regularizer $\mathcal{R}_k(\theta, \gamma)$. For such regularizer, we prove in section A.2 that (with a suitable choice of the learning rate $\lambda$) the aggregation step in (3) can be reduced to a weighted average of the local models.

The FedMAP algorithm consists of three main steps: Initialization, Local Optimization, and Global Aggregation, as outlined in Algorithm 1.

**Initialization**    A client $j$ is randomly selected, and its model parameters are used to initialize the global model $\gamma^{(0)}$ and the local model parameters $\theta_j^{(0)}$. These initial parameters are then broadcasted to all clients, ensuring that every client starts from a common initial point.

---

**Algorithm 1** FedMAP (Federated Maximum A Posteriori)

---

1: **Input:** $q$ (Total number of clients)
2: **Initialization:**
3: Randomly select client $j$ from $\{1, \ldots, q\}$
4: Initialize $\theta^{(0)}$ and $\gamma^{(0)}$ based on client $j$'s model
5: Broadcast $\gamma^{(0)}$ and $\theta^{(0)}$ to all clients
6: **for** each communication round $t = 0, 1, 2, \ldots$ **do**
7:     **for** $k = 1$ to $q$ **in parallel do**
8:         LOCALOPTIMIZATION($\gamma^{(t)}$)                                       ▷ Algorithm 2
9:     **end for**
10:        GLOBALAGGREGATION($\{\theta_k^{(t+1)}, \omega_k^{(t)}\}_{k=1}^q$)                ▷ Algorithm 3
11: **end for**

---

**Local Optimization**    Each client $k$ optimizes their model parameters $\theta_k^{(t+1)}$ by minimizing the negative log-likelihood of the posterior distribution. Using the explicit form of the prior in (6), we can write the minimization problem as

$$\theta_k^{(t+1)} = \arg\min_\theta \frac{1}{N_k} \sum_{i=1}^{N_k} \mathcal{L}\left(y_k^{(i)}, \phi(x_k^{(i)}; \theta)\right) + \frac{1}{2\sigma^2}\|\theta - \gamma^{(t)}\|^2, \tag{7}$$

where $\mathcal{L}\left(y_i, \phi(x_k^{(i)}; \theta)\right) := -\log \mathbb{P}\left(y_k^{(i)} | \phi(x_k^{(i)}, \theta)\right)$ denotes the loss function, and $N_k$ represents the count of data points in $Z_k$. The prior term penalizes deviations from the global model parameters $\gamma^{(t)}$. Each client iteratively updates $\theta_k^{(t)}$ for a fixed number of local epochs $e$, as detailed in Algorithm 2.

After optimizing the local model, each client computes a weighting factor $\omega_k^{(t)}$, representing the importance of the client's local model in the subsequent Global Aggregation step:

$$\omega_k^{(t)} = \mathbb{P}(Z_k | \theta_k^{(t+1)}) \times \rho_{\gamma^{(t)}}(\theta_k^{(t+1)}) \tag{8}$$

The locally optimized parameters $\theta_k^{(t+1)}$ and their corresponding weighting factor $\omega_k^{(t)}$ are then sent to the server for global aggregation.

---

**Algorithm 2** Local Optimization

---

1: **Input:**
2: $\theta^{(0)}$ The initial model parameter
3: $Z_k$**:** The local dataset for client $k$
4: $\gamma^{(t)}$**:** Current global model parameter
5: $\theta_k^{(t)}$**:** Current model parameters for client $i$
6: $e$**:** Number of epochs per local optimization
7: **if** $t = 0$ **then**
8:     $\theta_i^{(t)} \leftarrow \theta^{(0)}$
9: **end if**
10: **for** $epoch = 1$ to $e$ **do**
11:     $\theta_k^{(t+1)} \leftarrow \arg\min_\theta -\log \mathbb{P}(Z_k|\theta) - \log \rho_{\gamma^{(t)}}(\theta)$
12: **end for**
13: $\omega_k^{(t)} \leftarrow \mathbb{P}(Z_k|\theta_k^{(t+1)}) \times \rho_{\gamma^{(t)}}(\theta_k^{(t+1)})$
14: Send $\theta_k^{(t+1)}$ and $\omega_k^{(t)}$ to the server

---

**Global Aggregation**   The server aggregates the optimized local model parameters $\theta_k^{(t+1)}$ from all clients to obtain the updated global model parameters $\gamma^{(t+1)}$, as shown in Algorithm 3. In view of (3) and the specific form of $\rho_\gamma(\theta)$ in (6), the aggregation is performed as a weighted average of the local model parameters, where the weights are the weighting factors $\omega_k^{(t)}$ computed in the Local Optimization step:

$$\gamma^{(t+1)} = \frac{1}{\sum_{j=1}^q \omega_j^{(t)}} \sum_{k=1}^q \omega_k^{(t)} \theta_k^{(t+1)} \tag{9}$$

The updated global model $\gamma^{(t+1)}$ is then broadcast to all clients for the next round of Local Optimization.

Local Optimization and Global Aggregation steps are repeated iteratively until a predefined number of communication rounds is reached. After the iterative process is complete, the final personalized model for each client $k$ is given by the optimized local model parameters $\theta_k^{(t+1)}$. These personalized models capture the unique characteristics of each client's data while benefiting from the collective knowledge of all clients through the regularization effect of the global model.

---

**Algorithm 3** Global Aggregation

---

1: **Input:**
2: $q$**:** Total number of clients
3: $\theta_k^{(t+1)}$**:** Optimized model parameters from each client
4: $\omega_k^{(t)}$**:** Weighting factors from each client
5: $\gamma^{(t+1)} \leftarrow \dfrac{1}{\sum_{j=1}^N \omega_j^{(t)}} \sum_{k=1}^q \omega_k^{(t)} \theta_k^{(t+1)}$
6: Broadcast $\gamma^{(t+1)}$ to the all clients

---

## 4    Experiments

### 4.1    Datasets

To evaluate the performance of FedMAP under non-IID data distributions, we utilized both synthetic and public datasets.

Synthetic Datasets: A number of synthetic datasets were created to evaluate the FedMAP's ability to handle practical FL challenges. The evaluation scenarios are designed to reflect the complex combinations of non-IID issues that occur in practical applications. The datasets include three primary types of non-IID data distributions across clients: 1) Feature distribution skew, where each client's data is influenced by unique affine transformations. These transformations vary the feature space across clients; 2) Quantity Skew with each client has datasets in varying sample sizes; 3) Label Distribution Skew by different class proportions across the clients. The details of the synthetic data are provided in Appendix B.

Office-31 Dataset: To complement the experiments on synthetic datasets, we also used the Office-31 dataset (Saenko et al., 2010). This dataset consists of 4110 images across 31 object categories. The data are collected from three distinct domains: Amazon (images from Amazon.com), Webcam (low-resolution images captured by a webcam), and DSLR (high-resolution images captured by digital SLR cameras). The diversity in image domains, resolution, and acquisition conditions in the Office-31 dataset naturally introduces realistic non-IID data distributions.

### 4.2    Setup

In our experiments, we chose a Gaussian prior for the local model parameters $\theta$, defined by the probability density function (6), where $\gamma$ represents the global model parameters and $\sigma^2$ is the variance controlling the influence of the prior. When $\sigma \to 0$, the prior is centered at $\gamma$ and approximates FedAvg, prioritizing the global consensus. When $\sigma \to \infty$, the prior becomes a uniform distribution

over $\mathbb{R}^d$, removing any influence from the global model and allowing purely local learning, which maximizes client-specific adaptation. We selected this prior because the negative logarithm of the prior adds a convex quadratic term $\|\theta - \gamma\|^2/(2\sigma^2)$ to the local objective function, which makes the optimization efficient with gradient-based methods.

We evaluated FedMAP across three non-IID scenarios, and each scenario involved 10 clients for FL training and validation. In the first scenario, each client holds 2000 samples. In the second scenario, five clients hold 2000 samples each, and the other five hold only 500 samples each. In the third scenario, all clients have 2000 samples, but five clients have 85% of their samples belonging to class 0, while the class proportions were balanced for the rest of the clients. In all scenarios, each client employed a 70:30 train-validation split. The model architecture employed was a multi-layer perceptron (MLP) with two hidden layers. Models were optimized using the Adam optimizer. Detailed model architectures are provided in the Appendix B, and all other training details, including hyperparameters and settings for FL, can be found in the code repository: https://anonymous.4open.science/r/FedMAP-963/.

To simulate non-IID data distributions in the Office-31 dataset, we partitioned the data across three clients, each including a distinct domain: Amazon, Webcam, and DSLR. We utilized the Convolutional Neural Network (CNN) model architecture from the FedBN (Li et al., 2021b) experiments, which consists of five convolutional layers, each with batch normalization and ReLU activation, ending with an output layer for classification. Models were optimized using the SGD optimizer, and each client also used a 70:30 train-validation split.

FedMAP was evaluated against three established FL baselines and individual client training. FedAvg (McMahan et al., 2016) serves as the benchmark of FL. FedProx (Li et al., 2020) was selected due to its similarity to FedMAP in addressing non-IID by using the regularization approach. FedBN (Li et al., 2021b) was selected as it is a PFL method that targets similar non-IID scenarios as the FedMAP does. Individual client training, where models are trained and validated exclusively on each client's local data to assess base performance. All experiments were conducted on an AMD 5965WX 24-core CPU with two NVIDIA RTX A5500 GPUs. To facilitate reproducibility and practical adoption, we integrated FedMAP into one of the popular open-source frameworks, Flower (Beutel et al., 2020), enabling simulations of real distributed deployments.

## 4.3 RESULTS AND DISCUSSION

FedMAP shows prominent performance across non-IID scenarios, especially for clients with limited or imbalanced data. To ensure consistency and robustness of the results, we conducted each experimental setup 10 times and recorded the mean and standard deviation for these runs as our final results.

Table 1 illustrates the performance of FedMAP in the label distribution skew scenario, where clients 1-5 have balanced class distributions, whereas clients 6-10, marked in red, have significantly imbalanced class distributions. FedMAP is highly effective in this scenario, especially for clients 6-10 with severely imbalanced class distributions. Clients 8, 9, and 10 achieve accuracy improvements of over 13% compared to their individual training models. Notably, FedMAP also enhances the performance of the clients with balanced class distributions, demonstrating its ability to leverage diverse data to benefit all clients.

In contrast, existing FL methods FedAvg, FedProx, and FedBN underperform compared to individual client training across all clients with skewed label distributions. They show particularly large gaps in performance on the clients (6-10) with the most skewed data, highlighting their limitations in handling non-IID data distributions. For a comprehensive overview of FedMAP's performance in other non-IID scenarios, such as feature distribution and quantity skew, please refer to Tables 5 and 6 in the Appendix B.

As shown in Figure 1, the validation accuracy curves of clients using FedMAP vary significantly across different types of data skew. Under feature distribution skew (Figure 1a), all clients consistently achieve high validation accuracy and demonstrate stable learning trajectories, indicating FedMAP's effectiveness in mitigating feature distribution disparities. In contrast, clients under quantity and label distribution skews exhibit more variability and slower convergence.

Table 1: Accuracy comparison of FedMAP, FedBN, FedProx, FedAvg, and individual client training on the synthetic dataset with label distribution skew. Clients 1-5 have balanced label distributions, while 6-10 (red) have severely skewed label distributions. Standard deviations are shown below each accuracy value.

| Client | Individual | FedMAP | FedBN | FedProx | FedAvg |
|---|---|---|---|---|---|
| 1 | 87.32% | 89.03% (↑ 1.71%) | 67.11% | 55.64% | 55.30% |
|  | ±0.10% | ±0.08% | ±0.23% | ±0.20% | ±0.12% |
| 2 | 88.23% | 88.72% (↑ 0.49%) | 65.93% | 54.92% | 55.55% |
|  | ±0.11% | ±0.08% | ±0.33% | ±0.11% | ±0.11% |
| 3 | 89.94% | 90.92% (↑ 0.98%) | 69.52% | 56.11% | 55.54% |
|  | ±0.11% | ±0.05% | ±0.49% | ±0.42% | ±0.21% |
| 4 | 89.35% | 90.52% (↑ 1.17%) | 69.41% | 57.13% | 56.33% |
|  | ±0.14% | ±0.16% | ±0.54% | ±0.25% | ±0.38% |
| 5 | 86.96% | 88.01% (↑ 1.05%) | 68.65% | 56.50% | 56.57% |
|  | ±0.19% | ±0.07% | ±0.33% | ±0.29% | ±0.28% |
| 6 | 73.95% | 84.25% (↑ 10.30%) | 59.76% | 53.74% | 53.27% |
|  | ±0.92% | ±0.13% | ±0.62% | ±0.24% | ±0.32% |
| 7 | 63.86% | 79.37% (↑ 15.51%) | 56.85% | 53.68% | 54.68% |
|  | ±0.81% | ±0.18% | ±0.59% | ±0.22% | ±0.28% |
| 8 | 61.42% | 81.16% (↑ 19.74%) | 59.12% | 52.37% | 52.05% |
|  | ±0.67% | ±0.48% | ±0.38% | ±0.33% | ±0.24% |
| 9 | 61.02% | 80.52% (↑ 19.50%) | 55.02% | 53.18% | 53.34% |
|  | ±0.40% | ±0.33% | ±0.60% | ±0.22% | ±0.38% |
| 10 | 64.28% | 75.92% (↑ 11.64%) | 61.41% | 55.79% | 53.98% |
|  | ±0.29% | ±0.25% | ±0.39% | ±0.15% | ±0.22% |
| **Average** | 76.63% | 84.84% | 63.28% | 54.91% | 54.66% |
|  | ±0.37% | ±0.18% | ±0.45% | ±0.24% | ±0.25% |

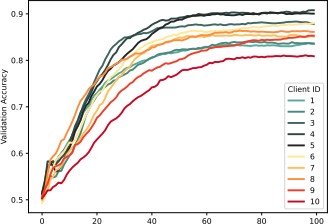 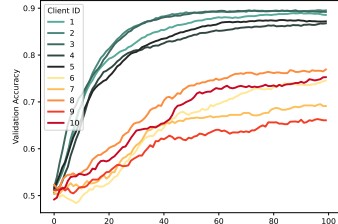 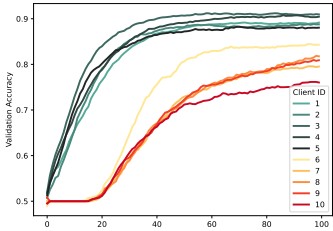

(a) Feature Distribution Skew: All clients have the same number of samples (2000 each).

(b) Quantity Skew: Clients 1-5 have 2000 samples each, while clients 6-10 have only 500 each.

(c) Label Distribution Skew: Clients 1-5 are balanced, while clients 6-10 have 85% of samples in one class.

Figure 1: Comparison of FedMAP's validation accuracy curves under three non-IID scenarios, illustrating the impact of feature, quantity, and label distribution skews on model performance.

In scenarios with quantity skew (Figure 1b), clients 6-10 initially show much lower accuracy than clients 1-5 due to their limited sample sizes. However, as communication rounds progress, these clients gradually improve, though slower.

For label distribution skew (Figure 1c), clients 6-10 experience an extended initial phase with low accuracy, indicating a strong bias towards the majority class. This phase persists for about 20 communication rounds which indicates the difficulty in learning the underrepresented minority class from severely imbalanced data distributions. Eventually, clients 6-10 converge, with their accuracy gradually improving. This transition suggests that the global prior distribution, shaped by the aggregation of local models, gradually adapts to the skewed label distributions, enabling clients 6-10 to learn the minority class better. Despite these challenges, FedMAP enhances overall validation accuracy, demonstrating robustness in heterogeneous data environments.

Table 2: Comparison of accuracies of all clients for each approach under Office-31 dataset.

| Domain | Individual | FedMAP | FedBN | FedProx | FedAvg |
|--------|-----------|--------|-------|---------|--------|
| Amazon | 65.80% | 70.34% (↑ 4.54%) | 70.29% | 64.37% | 64.86% |
| | ±0.87% | ±0.61% | ±0.30% | ±0.30% | ±0.19% |
| Webcam | 68.98% | 86.04% (↑ 17.06%) | 83.98% | 40.22% | 42.22% |
| | ±2.36% | ±0.20% | ±2.73% | ±1.77% | ±1.55% |
| DSLR | 85.57% | 95.54% (↑ 9.97%) | 82.52% | 75.52% | 77.49% |
| | ±2.38% | ±0.13% | ±2.47% | ±2.00% | ±0.87% |
| **Average** | 73.45% | 83.97% | 78.93% | 60.04% | 61.52% |
| | ±1.87% | ±0.32% | ±1.83% | ±1.36% | ±0.87% |

The results from the Office-31 dataset experiment further confirm FedMAP's effectiveness in handling non-IID data distributions encountered in real-world FL scenarios. As depicted in Table 2, FedMAP demonstrates performance gains over individual client training models across all three clients in the Office-31 dataset. Moreover, the varying degrees of improvement observed across clients can be attributed to the inherent diversity in data distributions present in the dataset. The Webcam domain shows the most significant accuracy gain, 17.06% with FedMAP. This can be explained by the low-resolution and unique characteristics of the Webcam images, which deviate substantially from the other domains. By leveraging the global prior, FedMAP effectively transfers relevant knowledge from the higher-resolution DSLR dataset, enabling the Webcam domain to overcome the limitations of its low-resolution data. Overall, the experiment results highlight FedMAP's ability to effectively address the challenges posed by non-IID data distributions.

## 5 CONCLUSION

In this paper, we proposed FedMAP, a novel FL framework incorporating a global prior distribution over local model parameters and enabling personalized FL. We formulated a mathematical framework for the problem with bi-level optimization, capturing the data heterogeneity across clients. Extensive evaluations across scenarios, including skewed feature, label and quantity distributions, have demonstrated FedMAP's performance gains over the existing methods such as FedAvg, FedProx, and FedBN. Additionally, the theoretical analysis positions FedMAP as a promising approach for robust, personalized federated learning in heterogeneous data environments. From the practical perspective, the Flower framework integration further improves reproducibility and practical deployment using Docker containers. It allows researchers and practitioners to incorporate FedMAP into their existing FL pipelines easily.

FedMAP can benefit healthcare particularly, as it encourages multiple hospitals to train ML models collaboratively despite their siloed data. Hospitals face non-IID issues as the variations in disease prevalence, demographic differences, and different dataset sizes (Zhang et al., 2024). FedMAP allows the training of robust and personalized diagnostic models tailored to each healthcare provider's unique data distribution, unlocking the potential of collaborative learning while preserving data privacy and integrity.

Although the results are promising, the limitations include assuming the global prior is an isotropic Gaussian distribution and the mean acts as a unique parameter. Therefore, future work will involve looking into issues such as covariance matrix tuning and exploring alternatives to Gaussian distributions for priors. We believe FedMAP offers an effective framework and streamlines collaboration in distributed data environments.

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

## A  CONVERGENCE ANALYSIS OF FEDMAP

We consider $q$ clients with datasets $Z_k \in (\mathcal{X} \times \mathcal{Y})^{N_k}$ for each $k = 1, \ldots, q$. As mentioned in section 2, in the FedMAP framework, each client obtains its local model by solving a regularized empirical risk minimization problem as follows:

$$\theta_k^{(t)} \in \arg\min_{\theta \in \Theta} \left\{ \mathcal{L}(\theta; Z_k) + \mathcal{R}(\theta, \gamma^{(t)}) \right\}, \tag{10}$$

where $\mathcal{L}(\theta; Z_k)$ is the empirical risk, and $\mathcal{R}(\theta, \gamma^{(t)})$ is a parametric regularizer, where the parameter is $\gamma^{(t)} \in \Gamma \subset \mathbb{R}^p$. If one considers a parametric regularizer $\mathcal{R}(\theta, \gamma)$ such that, for every $\gamma \in \Gamma$, the function $\theta \mapsto \mathcal{R}(\theta, \gamma)$ satisfies suitable growth assumptions, the optimization problem (10) can be viewed as a MAP estimator for the parametric prior distribution on $\Gamma$, with probability density given by

$$\rho_\gamma(\theta) = \frac{\exp(-\mathcal{R}(\theta, \gamma))}{C_\gamma}, \qquad \text{where} \quad C_\gamma = \int_\Theta \exp(-\mathcal{R}(\omega, \gamma)) \, d\omega.$$

Once each client has trained its local model $\theta_k^{(t)}$, this one is transmitted to the central node, that aggregates them to update the parameter $\gamma^{(t)}$ of the regularizer as follows:

$$\gamma^{(t+1)} = \gamma^{(t)} - \lambda \sum_{k=1}^q w_k \nabla_\gamma \mathcal{R}(\theta_k^{(t)}, \gamma^{(t)}), \tag{11}$$

where $\lambda > 0$ is the learning rate for the aggregation, and $(w_1, \ldots, w_q) \in (0, 1)^q$ with $\sum_k w_k = 1$ are weights than can be chosen based on the size of the datasets or the quality of the data of each client. Note that during the local training (10), the local data $Z_k$ appears only in the term $\mathcal{L}(\theta; Z_k)$, and therefore, no data needs to be shared during the aggregation step in (11).

## A.1 MAIN RESULT

Our main goal in this section is to prove that, under suitable convexity assumptions, the communication rounds in FedMAP, given by the local training (10) and the aggregation (11), correspond to gradient descent iterations of a strongly convex function $M(\gamma)$, defined as the linear combination of envelop functions associated with the local losses $\mathcal{L}(\theta; Z_k)$ and the parametric regularizer $\mathcal{R}(\theta, \gamma)$. Moreover, the minimizer of $M(\gamma)$ provides the unique solution to the bi-level optimization problem

$$
\begin{aligned}
&\underset{\theta_k \in \Theta}{\text{minimize}} \quad \mathcal{L}(\theta_k; Z_k) + \mathcal{R}(\theta_k, \gamma^*) \qquad \text{for each } k = 1, \dots, q, \\
&\text{s.t.} \quad \gamma^* \in \arg\min_{\gamma \in \Gamma} \left( \sum_{k=1}^{q} w_k \mathcal{R}(\theta_k, \gamma) \right).
\end{aligned}
\tag{12}
$$

Throughout this section, we shall make the following convexity assumption on $\mathcal{L}$ and $\mathcal{R}$.

**Assumption 1.** *The parameter space $\Theta \subset \mathbb{R}^d$ is compact and convex, and $\Gamma = \mathbb{R}^p$ for some $p \in \mathbb{N}$. Moreover, for each $k = 1, \dots, q$, the function $\theta \mapsto \mathcal{L}(\theta; Z_k)$ is continuous in $\Theta$ and convex, i.e. it satisfies*

$$
\frac{1}{2}\mathcal{L}(\theta_1; Z_k) + \frac{1}{2}\mathcal{L}(\theta_2; Z_k) \geq \mathcal{L}\left(\frac{\theta_1 + \theta_2}{2}; Z_k\right) \qquad \forall \theta_1, \theta_2 \in \Theta.
$$

*We also assume that the function $(\theta, \gamma) \mapsto \mathcal{R}(\theta, \gamma)$ is differentiable and strongly convex in $\Theta \times \Gamma$, i.e. there exists $\alpha > 0$ such that*

$$
\frac{1}{2}\mathcal{R}(\theta_1, \gamma_1) + \frac{1}{2}\mathcal{R}(\theta_2, \gamma_2) \geq \mathcal{R}\left(\frac{\theta_1 + \theta_2}{2}, \frac{\gamma_1 + \gamma_2}{2}\right) + \alpha \left( \left\|\frac{\theta_1 - \theta_2}{2}\right\|^2 + \left\|\frac{\gamma_1 - \gamma_2}{2}\right\|^2 \right),
$$

*for all $(\theta_1, \gamma_1), (\theta_2, \gamma_2) \in \Theta \times \Gamma$.*

For a given dataset $Z_k \in (\mathcal{X} \times \mathcal{Y})^{N_k}$ and a function $\mathcal{R} : \Theta \times \Gamma \to \mathbb{R}$ satisfying Assumption 1, let us define the function

$$
M_k(\gamma; Z_k) := \min_{\theta \in \Theta} \{\mathcal{L}(\theta; Z_k) + \mathcal{R}(\theta, \gamma)\}.
\tag{13}
$$

Assumption 1 implies that the minimizer in the right-hand-side of (13) is attained at a unique $\theta \in \Theta$ which depends on $\gamma$. Our main theoretical contribution reads as follows. This is a more detailed version of Theorem 1, presented in the main paper.

**Theorem 2.** *Let $Z_k \in (\mathcal{X} \times \mathcal{Y})^{N_k}$ with $k = 1, \dots, q$ be $q$ datasets, and assume that the functions $\theta \mapsto \mathcal{L}(\theta; Z_k)$ and $(\theta, \gamma) \mapsto \mathcal{R}(\theta, \gamma)$ satisfy Assumption 1. For any $\gamma^{(0)} \in \Gamma$ and $\lambda > 0$, the sequence $\{\gamma^{(t)}\}_{t \in \mathbb{N}}$ given by the FedMAP iterations (11)–(10) can be written as*

$$
\gamma^{(t+1)} = \gamma^{(t)} - \lambda \nabla_\gamma M(\gamma^{(t)}),
$$

*where the function $M : \Gamma \to \mathbb{R}$ is defined as*

$$
M(\gamma) := \sum_{k=1}^{q} w_k M_k(\gamma; Z_k),
$$

*where $M_k(\gamma; Z_k)$ is given by (13). Moreover, the function $M(\cdot)$ is strongly convex in $\Gamma$ and its unique minimizer $\gamma^*$ is such that $(\theta_1^*, \dots, \theta_q^*)$ given by*

$$
\theta_k^* \in \arg\min_{\theta \in \Theta} \{\mathcal{L}(\theta_k; Z_k) + \mathcal{R}(\theta_k, \gamma^*)\}, \qquad \text{for all } k = 1, \dots, q,
$$

*is the unique solution of the bi-level optimization problem (12).*

The proof of this theorem is given in section A.3. The above result implies that, since FedMAP iterations are gradient descent iterations of a strongly convex function, with a suitable choice of the learning rate $\lambda$, one can ensure that $\gamma^{(t)}$ converges to the unique minimizer of $M(\cdot)$ and, therefore, to the solution of (12).

**Remark 1.** *Let us point out some important differences between FedMAP and other FL approaches such as FedAvg (McMahan et al., 2016), FedProx (Li et al., 2020), FedBN (Li et al., 2021b) and Ditto (Li et al., 2021a). First of all, most FL approaches focus on finding a global global model, which is in*

*the same hypothesis set as the local models. Instead, in FedMAP one seeks for an optimal regularizer, in a potentially different hypothesis set. Note that the parameter space $\Theta$ for the local model and the parameter space $\Gamma$ for the regularizer might be different.*

*Another important difference is that, whereas most FL approaches focus on estimating the global model $\theta^* \in \Theta$ by minimizing a function of the form*

$$F(\theta) = \sum_{k=1}^{q} w_k \mathcal{L}(\theta; Z_k), \tag{14}$$

*which is a weighted average of the local loss functions $\mathcal{L}(\theta; Z_k)$, FedMAP minimizes a linear combination of functions $M_k(\gamma; Z_k)$ defined in (13). In particular, FedMAP minimizes the function*

$$M(\gamma) = \sum_{k=1}^{q} w_k M_k(\gamma; Z_k) = \sum_{k=1}^{q} w_k \min_{\theta \in \Theta} \left\{ \mathcal{L}(\theta; Z_k) + \mathcal{R}(\theta, \gamma) \right\}. \tag{15}$$

*As we show in the section A.2 below, for a special choice of $\mathcal{R}(\theta, \gamma)$, and when the data across the clients comes from the same distribution, minimizing $F(\theta)$ in (14) and minimizing $M(\gamma)$ in (15) are equivalent (or close to equivalent). However, when the datasets $Z_k$ come from different distributions, the minimizers of $F(\theta)$ and $M(\gamma)$ might be very different, even for the case of a quadratic regularizer of the form $\mathcal{R}(\theta, \gamma) = (\theta - \theta^*)^\top A(s)(\theta - \theta^*)$. This phenomenon is illustrated in section A.4 through a simple example using linear regression. Here, $A(s)$ is a parametric positive definite diagonal matrix with parameter $s \in \mathbb{R}^d$ and $\theta^* \in \Omega$. Note that the parameter $\gamma$ of the regularizer $\mathcal{R}(\theta, \gamma)$ is of the form $\gamma = (s, \theta^*) \in \mathbb{R}^d \times \Theta$.*

### A.2 QUADRATIC REGULARIZER

Let us consider the special case when $\Theta = \Gamma = \mathbb{R}^d$ and, for a diagonal $d \times d$ matrix $D_\sigma$ with $\sigma = (\sigma_1, \dots, \sigma_d) \in \mathbb{R}_*^d$ in the main diagonal, let us consider the regularizer $\mathcal{R}(\theta, \gamma)$ defined as

$$\mathcal{R}(\theta, \gamma) = \frac{\|D_\sigma(\theta - \gamma)\|^2}{2} = \sum_{i=1}^{d} \frac{\sigma_i^2}{2} (\theta_i - \gamma_i)^2. \tag{16}$$

In this case, the function $M_k(\gamma; Z_k)$ is given by

$$M_k(\gamma; Z_k) = \min_{\theta \in \mathbb{R}^d} \left\{ \mathcal{L}(\theta; Z_k) + \frac{\|D_\sigma(\theta - \gamma)\|^2}{2} \right\}. \tag{17}$$

This function is an anisotropic variant of the Moreau envelope of the function $\theta \mapsto \mathcal{L}(\theta; Z_k)$, and is well studied in the field of convex optimization. One of the main features of Moreau envelopes is that minimizing $M_k(\gamma; Z_k)$ over $\gamma$ is equivalent to minimizing $\mathcal{L}(\theta; Z_k)$ over $\theta$. Therefore if all the local loss functions $\mathcal{L}(\theta; Z_k)$ were equal, minimizing $F(\theta) = \sum_{k=1}^{q} w_k \mathcal{L}(\theta; Z_k)$ and minimizing $M(\gamma) = \sum_{k=1}^{q} w_k M(\gamma; Z_k)$ would produce the same global model. The same argument can be used when the datasets $Z_k$ are not equal but come from the same distribution, in which case $\mathcal{L}(\theta; Z_k) \approx \mathbb{E}_Z[\mathcal{L}(\theta; Z)]$ for all $k = 1, \dots, q$.

However, it is important to note that in the non-IID case, which is the case that interests us in this paper, the local loss functions $\mathcal{L}(\theta; Z_k)$ might be rather different across the clients. In this case, minimizing a linear combination of Moreau envelopes such as

$$M(\gamma) = \sum_{k=1}^{q} w_k M(\gamma; Z_k)$$

is not equivalent to minimizing the function

$$F(\theta) = \sum_{k=1}^{q} w_k \mathcal{L}(\theta; Z_k).$$

Indeed, we show in subsection A.4, through an example using linear regression, that minimizing $M(\gamma)$ and minimizing $F(\theta)$ can produce very different results. The use of Moreau envelopes in

the framework of personalized Federated Learning was already proposed in T Dinh et al. (2020). However, our approach is much more general that the one proposed in T Dinh et al. (2020), which can be seen as the special case in which the parametric regularizer is chosen with the form (16).

One of the key considerations when using a quadratic regularizer such as (16) is the choice of the hyperparameters $\sigma_1, \ldots, \sigma_d$, which are related to the variance of each parameter in $\theta = (\theta_1, \ldots, \theta_d)$ across the clients. When constructing a parametric model, it is in general difficult to know a priori which model parameters should be similar across the clients, and which parameters should vary. Using the flexibility of our approach described in Section 2, we can consider a parametric regularizer of the form

$$\mathcal{R}(\theta, \gamma) = \sum_{i=1}^{d} \alpha(s_i) \frac{(\theta_i - \mu_i)^2}{2}, \qquad \text{with} \quad \gamma = (s, \mu) \in \mathbb{R}^d \times \Theta \qquad (18)$$

where $\alpha : \mathbb{R} \mapsto \mathbb{R}^+$ is a function that has to be suitably chosen in a way that $\mathcal{R}(\theta, \gamma)$ satisfies the convexity condition in Assumption 1.

Similarly to $\mathcal{R}(\theta, \gamma)$ given in (16), the choice of the regularizer $\mathcal{R}(\theta, \gamma)$ in (18) corresponds to the assumption of a Gaussian prior in the parameter space $\Theta$. The advantage of the parametrization of $\mathcal{R}(\theta, \gamma)$ in (18), is that the variance of the Gaussian prior is a parameter which can be learned during the aggregation. The main challenge of using $\mathcal{R}(\theta, \gamma)$ as in (18) is that it must satisfy the convexity condition in Assumption 1.

The following result provides a suitable choice for function $\alpha(\cdot)$ that ensures that a parametric regularizer, similar to (18), fulfills Assumption 1.

**Lemma 1.** *Let $d \in \mathbb{R}$, and for any $c > 0$, define the function $\alpha : (-c, \infty) \to \mathbb{R}$ given by $\alpha(s) = \dfrac{1}{s + c}$. Then, for any $\varepsilon > 0$ and any bounded interval $I \subset (-c, \infty)$, the function*

$$\mathcal{R}(\theta, \mu, s) = \sum_{i=1}^{d} \alpha(s_i) \frac{(\theta_i - \mu_i)^2}{2} + \varepsilon(\|s\|^2 + \|\mu\|^2)$$

*is strongly convex in $\mathbb{R}^d \times \mathbb{R}^d \times I^d$.*

**Remark 2.** *The term $\varepsilon(\|s\|^2 + \|\mu\|^2)$ is only added to the regularizer in (18) to ensure strong convexity. However, we note that this term is only relevant during the aggregation, and it can be dropped during the local training since it does not depend on the local parameter $\theta$:*

$$\theta_k^{(t)} \in \arg\min_{\theta \in \Theta} \left\{ \mathcal{L}(\theta; Z_k) + \mathcal{R}(\theta, \mu, s) \right\} = \arg\min_{\theta \in \Theta} \left\{ \mathcal{L}(\theta; Z_k) + \sum_{i=1}^{d} \alpha(s_i) \frac{(\theta_i - \mu_i)^2}{2} \right\}.$$

In section A.4, we show, through a simple example for linear regression, how a parametric regularizer of the form (18) can be used to address a FL problem with heterogeneous data. Let us now prove Lemma 1.

*Proof.* To prove that $\mathcal{R}(\theta, \mu, s)$ is strongly convex, it is enough to prove that the Hessian matrix is definite positive for all $(\theta, \mu, s) \in \Theta \times \Theta \times I^d$, and that the smallest eigenvalue can be bounded away from 0 independently of $(\theta, \mu, s)$. The function $\mathcal{R}(\theta, \mu, s)$ has $3d$ variables, and therefore, its Hessian matrix is of size $3d \times 3d$. However, since $\mathcal{R}(\theta, \mu, s)$ can be written as the sum of $d$ terms in the following way:

$$\mathcal{R}(\theta, \mu, s) = \sum_{i=1}^{d} \left( \alpha(s_i) \frac{(\theta_i - \mu_i)^2}{2} + \varepsilon s_i^2 + \varepsilon \mu_i^2 \right) = \sum_{i=1}^{d} F(\theta_i, \mu_i, s_i),$$

the Hessian matrix of $\mathcal{R}(\theta, \mu, s)$ is block diagonal with $d$ blocks of size $3 \times 3$ of the form

$$H_i = \begin{bmatrix} F_{\theta\theta}(\theta_i, \mu_i, s_i) & F_{\theta\mu}(\theta_i, \mu_i, s_i) & F_{\theta s}(\theta_i, \mu_i, s_i) \\ F_{\theta\mu}(\theta_i, \mu_i, s_i) & F_{\mu\mu}(\theta_i, \mu_i, s_i) & F_{\mu s}(\theta_i, \mu_i, s_i) \\ F_{\theta s}(\theta_i, \mu_i, s_i) & F_{\mu s}(\theta_i, \mu_i, s_i) & F_{ss}(\theta_i, \mu_i, s_i), \end{bmatrix}$$

where the sub-indexes represent the partial derivatives of $F$. Proving that the Hessian matrix of $\mathcal{R}(\theta, \mu, s)$ is definite positive is equivalent to proving that each block $H_i$ is definite positive.

From now on, we omit the dependence of $F$ on $(\theta_i, \mu_i, s_i)$ to make the notation lighter. Using simple calculus, one can compute the second derivatives in $H_i$ as

$$F_{\theta\theta} = 2\alpha(s_i), \qquad F_{\theta\mu} = -2\alpha(s_i), \qquad F_{\theta s} = 2\alpha'(s_i)(\theta_i - \mu_i),$$

$$F_{\mu\mu} = 2\alpha(s_i) + \varepsilon, \qquad F_{\mu s} = -2\alpha'(s_i)(\theta_i - \mu_i), \qquad F_{ss} = \alpha''(s)(\theta_i - \mu_i)^2 + \varepsilon.$$

Since $H_i$ is a symmetric matrix, it is well-known, by Sylvester's criterion, that $H_i$ is definite positive if and only if all its leading principal minors are definite positive.

The leading principal minor of order one is simply the scalar $F_{\theta\theta} = 2\alpha(s_i) = \dfrac{2}{s_i + c} > 0$ which is uniformly positive in the bounded interval $I$. The leading principal minor of order two is

$$\begin{vmatrix} 2\alpha(s_i) & -2\alpha(s_i) \\ -2\alpha(s_i) & 2\alpha(s_i) + \varepsilon \end{vmatrix} = 2\varepsilon\alpha(s_i) = \frac{2}{s_i + c} > 0,$$

which, again, is uniformly positive in the bounded interval $I$. After some computations, one can verify that the leading principal minor of order three can be written as

$$M_3 = \begin{vmatrix} 2\alpha(s_i) & -2\alpha(s_i) & 2\alpha'(s_i)(\theta_i - \mu_i) \\ -2\alpha(s_i) & 2\alpha(s_i) + \varepsilon & -2\alpha'(s_i)(\theta_i - \mu_i) \\ 2\alpha'(s_i)(\theta_i - \mu_i) & -2\alpha'(s_i)(\theta_i - \mu_i) & \alpha''(s_i)(\theta_i - \mu_i)^2 + \varepsilon \end{vmatrix}$$

$$= 2\varepsilon(\theta_i - \mu_i)\left(\alpha(s_i)\alpha''(s_i) - 2(\alpha'(s_i))^2\right) + 2\varepsilon^2\alpha(s_i).$$

We can see that the function $\alpha(s) = \dfrac{1}{s + c}$ satisfies $\alpha(s)\alpha''(s) = 2(\alpha'(s))^2$ for all $s > -c$. Hence, the leading principal minor of order three is given by $M_3 = 2\varepsilon^2\alpha(s_i) > 0$ which is uniformly positive in $I$. This implies that each block $H_i$ in the Hessian of $\mathcal{R}(\theta, \mu, s)$ is uniformly positive in $\Theta \times \Theta \times I^d$, and hence, we conclude that the function $\mathcal{R}(\theta, \mu, s)$ is strongly convex in $\Theta \times \Theta \times I^d$. $\qquad\square$

### A.3 Proof of Theorem 2

*Proof.* For each $k \in \{1, \ldots, q\}$ and for any $\gamma \in \Gamma$, by the Assumption 1, the function $\theta \mapsto \mathcal{L}(\theta; Z_k) + \mathcal{R}(\theta, \gamma)$ is strongly convex in $\Theta$, and therefore, there exists a unique $\theta_k^* \in \Theta$ such that

$$M_k(\gamma; Z_k) = \mathcal{L}(\theta_k^*; Z_k) + \mathcal{R}(\theta_k^*, \gamma).$$

Now, by means of Danskin's Theorem (see (Danskin, 1966; Bernhard & Rapaport, 1995)), and since the loss $\mathcal{L}(\theta; Z_k)$ is independent of $\gamma$, the gradient of $M_k(\cdot; Z_k)$ at $\gamma$ is given by

$$\nabla_\gamma M_k(\gamma; Z_k) = \nabla_\gamma \mathcal{R}(\theta_k^*, \gamma).$$

Since this is true for any $k \in \{1, \ldots, q\}$, we obtain

$$\nabla_\gamma M(\gamma) = \sum_{k=1}^{q} w_k \nabla_\gamma M_k(\gamma; Z_k) = \sum_{k=1}^{q} w_k \nabla_\gamma \mathcal{R}(\theta_k^*, \gamma),$$

where $\theta_k^*$ is the unique minimizer in $\Theta$ of $\theta \mapsto \mathcal{L}(\theta; Z_k) + \mathcal{R}(\theta, \gamma)$. Hence, we can write the update formula (11) as

$$\gamma^{(t+1)} = \gamma^{(t)} - \lambda \nabla_\gamma M(\gamma^{(t)}).$$

The strong convexity of $M(\gamma)$ is proved in Lemma 2 below.

Let us now prove that the minimizer of $M(\gamma)$, denoted by $\gamma^*$, together with $(\theta_1^*, \ldots, \theta_q^*) \in \Theta^q$, obtained as the unique solution of the optimization problem

$$\underset{\theta \in \Theta}{\text{minimize}} \left\{ \mathcal{L}(\theta; Z_k) + \mathcal{R}(\theta, \gamma^*) \right\}, \qquad \text{for each } k = 1, \ldots, q,$$

is the unique solution to the bi-level optimization problem (12). For that, we will first prove that (12) has at least one solution, and then we will prove that, for any solution $(\theta_1^*, \ldots, \theta_q^*)$ of (12), the unique

$\gamma^* \in \Gamma$ satisfying $\gamma^* \in \arg\min_\gamma \sum_{k=1}^q w_k \mathcal{R}(\theta_k^*, \gamma^*)$ is a critical point of $M(\gamma)$. The uniqueness of the solution follows from the strong convexity of $M(\gamma)$.

By the compactness of $\Theta$ and the strong convexity of $\mathcal{R}(\theta, \gamma)$, it follows that the function $(\theta, \gamma) \mapsto \mathcal{L}(\theta; Z_k) + \mathcal{R}(\theta_k, \gamma^*)$ is bounded from below for all $k = 1, \ldots, q$. Then, there exists a minimizing sequence $\{(\theta_1^{(n)}, \ldots, \theta_q^{(n)})\}_{n \in \mathbb{N}}$ in $\Theta^q$ associated to the minimization problem (12). By the compactness of $\Theta^q$, the minimizing sequence converges (through a subsequence) to some $(\theta_1^*, \ldots, \theta_q^*) \in \Theta^q$. Also, by the continuity and the strong convexity of the function $\gamma \mapsto \sum_{k=1}^q w_k \mathcal{R}(\theta_k, \gamma)$, there exists a unique sequence $\{\gamma_n^*\}_{n \in \mathbb{N}}$ in $\Gamma$, satisfying

$$\gamma_n^* \in \arg\min_\gamma \sum_{k=1}^q w_k \mathcal{R}(\theta_k^{(n)}, \gamma), \qquad \forall n \in \mathbb{N},$$

which also converges to the parameter $\gamma^* \in \Gamma$ associated to the limit point $(\theta_1^*, \ldots, \theta_q^*)$. Due to the continuity of $\theta \mapsto \mathcal{L}(\theta; Z_k)$ and $(\theta, \gamma) \mapsto \mathcal{R}(\theta, \gamma)$ we conclude that $(\theta_1^*, \ldots, \theta_q^*)$, together with $\gamma^*$ is a solution to the optimization problem (12).

Let us now prove that the parameter $\gamma^*$ associated to any solution of (12) is a critical point of $M(\gamma)$. By the first-order optimality condition, the parameter $\gamma^*$ associated to any solution $(\theta_1^*, \ldots, \theta_q^*)$ of (12) satisfies

$$\sum_{k=1}^q w_k \nabla_\gamma \mathcal{R}(\theta_k^*, \gamma^*) = 0.$$

Since for each $k \in \{1, \ldots, q\}$, $\theta_k^*$ is the unique minimizer of $\theta \mapsto \mathcal{L}(\theta; Z_k) + \mathcal{R}(\theta, \gamma^*)$, it is easy to deduce that $\gamma^*$ is a fixed point for the FedMAP iterations defined by (10)–(11), and then, since these iterations correspond to gradient descent iterations for the strongly convex function $M(\gamma)$, we deduce that $\gamma^*$ is the unique critical point of $M(\gamma)$. $\qquad\square$

In the next lemma we prove that the function $M(\gamma)$ defined in Theorem 2 is strongly convex in $\Gamma$.

**Lemma 2.** *Let $Z_k \in (\mathcal{X} \times \mathcal{Y})^{N_k}$ with $k = 1, \ldots, q$ be $q$ datasets, and consider that the loss functions $\mathcal{L} : \Theta \times \bigcup_{N \in \mathbb{N}} (\mathcal{X} \times \mathcal{Y})^N \to \mathbb{R}$ and $\mathcal{R} : \Theta \times \Gamma \to \mathbb{R}$ satisfy Assumption 1. Then, the function $M(\gamma)$ defined in Theorem 2 is strongly convex in $\Gamma$, i.e. it satisfies*

$$\frac{1}{2} M(\gamma_1) + \frac{1}{2} M(\gamma_2) \geq M\left(\frac{\gamma_1 + \gamma_2}{2}\right) + \alpha \left\|\frac{\gamma_1 - \gamma_2}{2}\right\|^2, \qquad \forall \gamma_1, \gamma_2 \in \Gamma.$$

*Proof.* Let $\gamma_1, \gamma_2 \in \Gamma$, and for each $k = 1, \ldots q$, let $\theta_{k,1}^*, \theta_{k,2}^* \in \Theta$ be such that

$$M_k(\gamma_i; Z_k) = \mathcal{L}(\theta_{k,i}^*; Z_k) + \mathcal{R}(\theta_{k,i}^*, \gamma_i), \qquad \text{for } i = 1, 2.$$

Using the definition of $M_k(\gamma; Z_k)$ in (13) and the Assumption 1, we obtain

$$
\begin{aligned}
M_k\left(\frac{\gamma_1 + \gamma_2}{2}; Z_k\right) &\leq \mathcal{L}\left(\frac{\theta_{k,1}^* + \theta_{k,2}^*}{2}; Z_k\right) + \mathcal{R}\left(\frac{\theta_{k,1}^* + \theta_{k,2}^*}{2}, \frac{\gamma_1 + \gamma_2}{2}\right) \\
&\leq \frac{1}{2}\mathcal{L}(\theta_{k,1}^*; Z_k) + \frac{1}{2}\mathcal{L}(\theta_{k,2}^*; Z_k) + \frac{1}{2}\mathcal{R}(\theta_{k,1}^*, \gamma_1) + \frac{1}{2}\mathcal{R}(\theta_{k,2}^*, \gamma_2) \\
&\quad -\alpha\left(\left\|\frac{\theta_{k,1}^* - \theta_{k,2}^*}{2}\right\|^2 + \left\|\frac{\gamma_1 - \gamma_2}{2}\right\|^2\right) \\
&\leq \frac{1}{2}M_k(\gamma_1; Z_k) + \frac{1}{2}M_k(\gamma_2; Z_k) - \alpha\left\|\frac{\gamma_1 - \gamma_2}{2}\right\|^2 \qquad \forall k = 1, \ldots q.
\end{aligned}
$$

Re-arranging the terms in the above inequality, we obtain

$$\frac{1}{2}M_k(\gamma_1; Z_k) + \frac{1}{2}M_k(\gamma_2; Z_k) \geq M_k\left(\frac{\gamma_1 + \gamma_2}{2}; Z_k\right) + \alpha\left\|\frac{\gamma_1 - \gamma_2}{2}\right\|^2 \qquad \forall k = 1, \ldots q,$$

and taking the summation over $k = 1, \ldots, q$, we conclude the proof. $\qquad\square$

## A.4 Example for linear regression

We consider a simple linear regression task, in which the goal is to estimate the parameters $(a, b) \in \mathbb{R}^2$ in the model

$$Y = aX + b + \varepsilon, \tag{19}$$

where $X \in \mathbb{R}$ is the input and $\varepsilon \in \mathbb{R}$ is Gaussian noise $N(0, \sigma^2)$, with variance $\sigma^2 = 0.8$. Let $Z_k = \{(x_k^{(i)}, y_k^{(i)})\}_{i=1}^{N_k}$, for $k = 1, \ldots, 5$, be $q = 5$ datasets. Since we want to address a heterogeneous setup, we assume that the parameter $b$ in $(a, b)$ is different for each dataset, as well as the distribution of the input variable $X$. We will also assume different sizes $N_k$ for the 5 datasets. More precisely, we consider

$$a = -1 \qquad \text{and} \qquad b_k = 4(k-1), \quad \text{for } k = 1, \ldots, 5.$$

Concerning the distribution of the input data, we consider that

$$X_k \sim N(k-1, 1), \qquad \text{for } k = 1, \ldots, 5.$$

In other words, the input $x_k^{(i)}$ in each data point $(x_k^{(i)}, y_k^{(i)})$ in $Z_k$ follows a Normal distribution with mean $k - 1$ and variance 1, and the output $y_k^{(i)}$ follows a Normal distribution with mean $ax_k^{(i)} + 4(k-1)$ and variance $\sigma^2 = 0.8$. As for the size of the datasets, we considered

$$(N_1, N_2, N_3, N_4, N_5) = (60, 1, 2, 3, 50).$$

See Figure 2a for a representation of the 5 datasets.

Since we are considering Gaussian noise in the model (19), a suitable choice for the local loss would be

$$\mathcal{L}(a, b; Z_k) = \frac{1}{N_k} \sum_{i=1}^{N_k} \left( ax_k^{(i)} + b - y_k^{(i)} \right)^2.$$

We recall that in this case, the parameter of the model is of the form $\theta = (a, b) \in \mathbb{R}^2$ and the dataset is given by $Z_k = \{(x_k^{(i)}, y_k^{(i)})\}_{i=1}^{N_k}$. Minimizing this loss function gives the MLE for the local model $k$, but shares no information with the other clients. This approach would be effective for clients 1 and 5, which have large datasets, be might result in poor performance for clients 2, 3 and 4, which have smaller datasets.

In many FL approaches, a global model $(a^*, b^*) \in \mathbb{R}^2$ is obtained by minimizing a function of the form

$$(a^*, b^*) \in \arg \min_{(a,b) \in \mathbb{R}^2} F(a, b) := \sum_{k=1}^{q} \frac{N_k}{N} \mathcal{L}(a, b; Z_k) = \frac{1}{N} \sum_{k=1}^{q} \sum_{i=1}^{N_k} \left( ax_k^{(i)} + b - y_k^{(i)} \right)^2. \tag{20}$$

See Figure 2b for a representation of the global model $(a^*, b^*)$ obtained by minimizing such function. Minimizing this function gives the MLE estimator for the union of the datasets $Z = \bigcup_{k=1}^{q} Z_k$. However, it ignores the fact that data points come from different clients. This can have a catastrophic effect for personalization purposes. Indeed, note that the real parameter $a$ is $-1$ for all the clients, whereas $a^*$ in the global model is positive.

Let us now consider the FedMAP approach with $\Theta = \mathbb{R}^2$ and $\Gamma = \mathbb{R}^2 \times (-1, \infty)^2$ and the parametric regularizer

$$\mathcal{R}_k(a, b, \mu_a, \mu_b, s_a, s_b) = \frac{\alpha(s_a)}{2}(a - \mu_a)^2 + \frac{\alpha(s_b)}{2}(b - \mu_b)^2 + \varepsilon(\mu_a^2 + \mu_b^2 + s_a^2 + s_b^2), \tag{21}$$

with $\varepsilon = 10^{-4}$ and $\alpha(s) = \dfrac{1}{s+1}$. This choice corresponds to a quadratic regularizer with the perturbation $\varepsilon(\mu_a^2 + \mu_b^2 + s_a^2 + s_b^2)$. The parameters of this regularizer are $(\mu_a, \mu_b, s_a, s_b) \in \Gamma$. The parameters $(\mu_a, \mu_b)$ represent the mean of the parameters $a$ and $b$, respectively, in the linear regression model, whereas the parameters $(s_a, s_b)$ are associated with the variance of these parameters. Note that utilizing a different variance for $a$ and $b$ is important since the parameter $a$ is the same in all the datasets, and the parameter $b$ varies across the different clients. Ideally, one should take $\alpha(s_a)$ much bigger than $\alpha(s_b)$, however, we do not assume that we have this information a priori. In the FedMAP approach, $\alpha(s_a)$ and $\alpha(s_b)$ are learned, through the parameters $s_a$ and $s_b$, during the aggregation

steps, according to the trained local models. The choice of the function $\alpha : (-1, \infty) \to \mathbb{R}^+$ is motivated by Lemma 1, ensuring that the convexity assumption of Theorem 2 is satisfied.

As shown in section A.1, the FedMAP iterations correspond to gradient descent applied to the function

$$M(\mu_a, \mu_b, s_a, s_b) = \sum_{k=1}^{q} w_k M_k(\mu_a, \mu_b, s_a, s_b; Z_k), \tag{22}$$

where the weights $w_k$ are chosen, based on the sample size of each dataset, as $w_k = N_k/N$, and $M_k(\mu_a, \mu_b, s_a, s_b; Z_k)$ is the envelope function given by

$$M_k(\mu_a, \mu_b, s_a, s_b; Z_k) = \min_{(a,b) \in \mathbb{R}^2} \{\mathcal{L}(a, b; Z_k) + \mathcal{R}_k(a, b, \mu_a, \mu_b, s_a, s_b)\},$$

where $\mathcal{R}_k(a, b, \mu_a, \mu_b, s_a, s_b)$ is the parametric regularizer defined in (21). Note that, by the form of the regularizer, we can write

$$M_k(\mu_a, \mu_b, s_a, s_b; Z_k) = \min_{(a,b) \in \mathbb{R}^2} \left\{ \mathcal{L}(a, b; Z_k) + \frac{\alpha(s_a)}{2}(a - \mu_a)^2 + \frac{\alpha(s_b)}{2}(b - \mu_b)^2 \right\}$$
$$+ \varepsilon(\mu_a^2 + \mu_b^2 + s_a^2 + s_b^2).$$

This implies that the perturbation $\varepsilon(\mu_a^2 + \mu_b^2 + s_a^2 + s_b^2)$ does not need to be considered during the local training.

We initialized the parameters $(\mu_a, \mu_b)$ following a Normal distribution $N(0, 1)$, and the parameters $(s_a, s_b)$ were initialised as $(0, 0)$, which corresponds to $\alpha(s_a) = \alpha(s_b) = 1$. After 10 communication rounds of FedMAP algorithm, we obtained the following parameters for the regularizer:

$$\mu_a = -1.0735 \quad \mu_b = 9.4882$$
$$s_a = 1.9431 \quad s_b = 97.7605$$

which after applying $\alpha$ to the parameters $s_a$ and $s_b$ yields $\alpha(s_a) = 0.3400$ and $\alpha(s_b) = 0.0101$. We observe that the estimated mean for the parameter $a$ is $\mu_a \approx -1$, and the estimated parameter $\alpha(s_a)$ is much larger than $\alpha(s_b)$. This implies that, after only 10 communication rounds, the estimated variance for the parameter $a$ is much smaller than that for the parameter $b$.

We see in Figure 2c that the linear regression model associated to the parameters $(a, b) = (\mu_a, \mu_b)$ obtained by minimizing $M(\mu_a, \mu_b, s_a, s_b)$ differs a lot from the one obtained when minimizing $F(a, b)$ in (20).

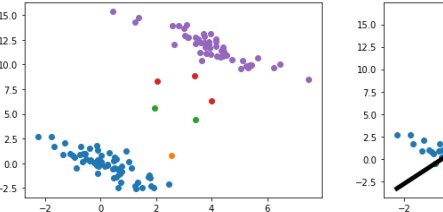 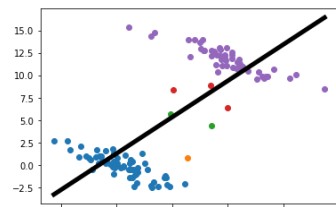 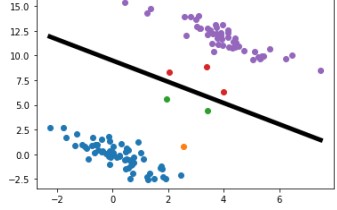

(a) Training data from the 5 datasets used in the example from A.4.  (b) Global model obtained by minimizing $F(a, b)$ in (20).  (c) Global model associated to $(a, b) = (\mu_a, \mu_b)$.

Figure 2: Training data and global models for the linear regression example from section A.4. The global models were obtained by minimizing two loss functions that combine the data across the clients differently. In (b), the parameters $(a, b)$ are obtained by minimizing $F(a, b)$ in 20, whereas in (c), we used $(a, b) = (\mu_a, \mu_b)$ where $(\mu_a, \mu_b)$ are obtained by minimizing $M(\mu_a, \mu_b, s_a, s_b)$ in (22).

Of course, none of the global models $(a^*, b^*)$ and $(\mu_a, \mu_b)$ can be reliably used to make predictions for the 5 clients. Therefore, in such a non-IID setting, a personalized FL approach needs to be implemented. The main idea in the Ditto approach Li et al. (2021a), as well as in the FedMAP approach, is to use a global model as a regularization term to train local models for each client. In the Ditto approach, the global model is obtained by minimizing the functional $F(a, b)$ in (20), and thus, the personalised models are obtained by minimizing the following functional:

$$(a_k, b_k) \in \arg \min_{(a,b) \in \mathbb{R}^2} \mathcal{L}(a, b; Z_k) + \frac{\sigma_a}{2}(a - a^*)^2 + \frac{\sigma_b}{2}(b - b^*)^2, \quad \text{for each } k = 1, \ldots, 5,$$

where $(a^*, b^*) \in \mathbb{R}^2$ is the solution to (20). The choice of the hyperparameters $\sigma_a$ and $\sigma_b$ in this approach is critical, and in general, it is not straightforward to make this choice without having prior knowledge about the datasets. In Figure 3b we see the 5 personalized models obtained by minimizing the above functional. To better visualize the mismatch of some of the local models with the local data, we plotted a larger dataset that has only been generated for test purposes. We observe that using $(a^*, b^*)$ in the regularisation term has a negative impact. This is due to the fact that the global model has been trained without taking into account the heterogeneity of the data.

Instead, the local models in FedMAP are obtained by minimizing the functional

$$(a_k, b_k) \in \arg\min_{(a,b)\in\mathbb{R}^2} \mathcal{L}(a,b; Z_k) + \frac{\alpha(s_a)}{2}(a - \mu_a)^2 + \frac{\alpha(s_b)}{2}(b - \mu_b)^2, \qquad \text{for each } k = 1, \ldots, 5,$$

where $(\mu_a, \mu_b, s_a, s_b) \in \Gamma$ are obtained by minimizing the functional $M(\mu_a, \mu_b, s_a, s_b)$ in (22). We can see in Figure 3c that using $(\gamma_a, \gamma_b)$ in the regularisation term produces much better personalized models. We also stress that, in this approach, the parameters $\alpha(s_a)$ and $\alpha(s_b)$ in front of the quadratic terms of the regularizer are not manually chosen. Instead, they are learned, during the aggregation steps, according to the local models.

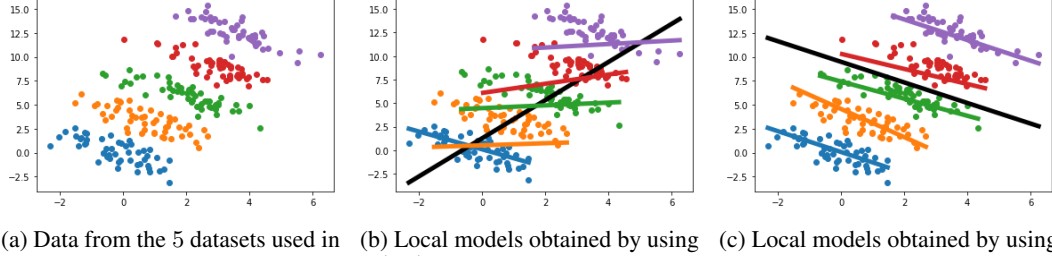

(a) Data from the 5 datasets used in A.4. This data was generated for test purposes only.

(b) Local models obtained by using $(a^*, b^*)$ and $(\sigma_a, \sigma_b)$ in the regularisation term.

(c) Local models obtained by using $(\mu_a, \mu_b)$ and $(\alpha(s_a), \alpha(s_b))$ in the regularisation term.

Figure 3: Test data from the example in section A.4 and local models trained through regularised empirical risk minimization. The data was generated for test purposes only. For the central plot, the parameters in the quadratic regularizer are $\sigma_a = \sigma_b = 0.1$ and $(a^*, b^*)$ obtained by minimizing $F(a, b)$ in (20). In the right plot, the parameters in the quadratic regularizer are $(\alpha(s_a), \alpha(s_b))$ and $(\mu_a, \mu_b)$ obtained by minimizing $M(\mu_a, \mu_b, s_a, s_b)$ in (22) respectively.

## B    SUPPLEMENTARY MATERIAL FOR THE EXPERIMENTS

### B.1    DESCRIPTION OF THE SYNTHETIC DATA

We designed a data synthesis method that generates synthetic data for a binary classification problem in $\mathbb{R}^n$, where $n = 30$ is the number of features. We assume that the intrinsic dimension of the data is $d = 4$, so we start by randomly selecting a $d$-dimensional linear subspace of $\mathbb{R}^n$. This is done by randomly generating $d$ orthonormal vectors of $\mathbb{R}^n$, that we stack as a matrix $B = (b_1, \ldots, b_d) \in \mathbb{R}^{n \times d}$. For every data point $(x, y) \in \mathbb{R}^n \times \{0, 1\}$, the label $y$ is correlated only with the projection of $x$ onto the space spanned by $B$, and the orthogonal component to $B$ is assumed to be a nuisance variable.

The data generation for the two classes is as follows:

1. For the label $y = 0$, the projection of $x$ onto the $d$-dimensional subspace $B$ follows an isotropic Normal distribution centered at the origin, i.e. $Bx \sim \mathcal{N}(\mathbf{0}, \sigma_0^2 \mathbf{I}_d)$, with $\sigma_0^2 = 2$.

2. For the label $y = 1$, the projection of $x$ onto the $d$-dimensional subspace $B$ is distributed around the $(d-1)$-dimensional sphere of radius 8 as follows: $\frac{Bx}{\|Bx\|} \sim \mathcal{U}(\mathbb{S}^{d-1})$, where $\mathcal{U}(\mathbb{S}^{d-1})$ denotes the uniform distribution on the unit $(d-1)$-dimensional sphere, and the norm of $Bx$ is normally distributed around 8, i.e. $\|Bx\| \sim \mathcal{N}(8, \sigma_1^2 \mathbf{I}_d)$, with $\sigma_1^2 = 2$.

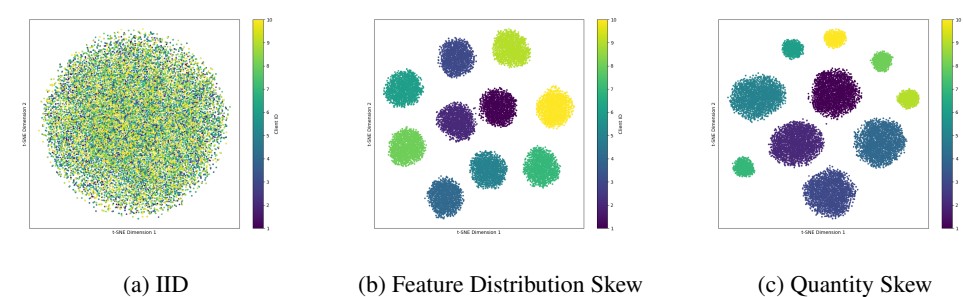

(a) IID                 (b) Feature Distribution Skew                 (c) Quantity Skew

Figure 4: t-SNE visualizations of synthetic data under different skew scenarios. The IID scenario (left) shows uniform data distribution across clients, while Feature Distribution Skew (middle) and Quantity Skew (right) demonstrate different types of non-IID scenarios.

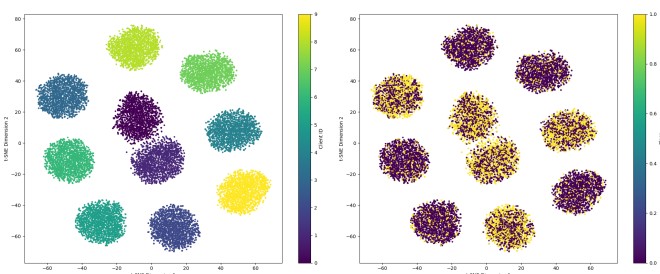

Figure 5: Label Distribution Skew

In both cases, the projection of the vector of features onto the orthogonal space to $B$ follows a Normal distribution $\mathcal{N}(\mathbf{0}, \sigma_0^2 \mathbf{I}_{n-d})$.

In this way, the binary classification task consists of separating the feature vectors $x$ such that $Bx$ is close to 0 from those such that $Bx$ is close to the $(d-1)$- dimensional sphere of radius 8, where $B = (b_1, \ldots, b_d)$ is an unknown $n \times d$ matrix with orthonormal columns.

This data generation process is repeated independently for each of the $q = 10$ clients, with the number of samples per machine determined by a split ratio. Finally, in order to simulate Feature Distribution Skew across the clients, for each client, the vector of features $x$ is further transformed by an affine transformation $Ax + b$, where $\mathbf{A} \in \mathbb{R}^{n \times n}$ is a random matrix and $\mathbf{b} \in \mathbb{R}^n$ is a random vector that scales with the client index. The matrix $A$ and the vector $b$ are different for each client.

The method introduces three types of data skew to simulate realistic FL non-IID scenarios. **Feature Distribution Skew** is achieved by applying client-specific random affine transformations $\mathbf{x} \mapsto A\mathbf{x}+\mathbf{b}$ to the local data points, with $A \in \mathbb{R}^{n \times n}$ and $\mathbf{b} \in \mathbb{R}^n$ varying between clients. This results in the data distribution shifts across clients. **Label Distribution Skew** and **Quantity Skew** are simulated by varying the proportion of class 0 samples and the number of samples assigned to each client, respectively, on top of Feature Distribution Skew. This results in clients having different class proportions, feature distributions, and numbers of samples.

The method allows fine-grained control over the degree and nature of the induced data skew, enabling the generation of realistic synthetic federated datasets to study the impact of different non-IID data scenarios on the performance of FL algorithms.

To verify and gain an in-depth understanding of our synthetic data generation process for different non-IID scenarios and compare them to IID scenarios, we employed t-SNE (t-distributed Stochastic Neighbor Embedding) (van der Maaten & Hinton, 2008) to reduce the high-dimensional data to 2D representations. The t-SNE visualizations, as shown in Figures 4 and 5, illustrate the different data heterogeneity scenarios in the datasets used in the evaluation. The IID scenario (Figure 4a) shows the idealistic data distribution for FL as it has no distribution skewness, with data points uniformly mixed across clients. The Feature Distribution Skew (Figure 4b) indicates distinct clusters for each

client, which confirms each client has a specific data distribution. The Quantity Skew (Figure 4c) shows the varying densities of points per client, which indicates imbalanced sample sizes. The Label Distribution Skew (Figure 5) illustrates the distribution variation in class. These visualizations provide us invaluable insights into the diverse challenges in non-IID data.

## B.2 Model architecture and training details on benchmark

For the synthetic dataset, we employed an MLP with two hidden layers (see Table 3). The simplicity of the MLP allows us to isolate and analyze the impact of non-IID on FedMAP's bi-level optimization interactions.

For the Office-31 dataset, we used the same Convolutional Neural Network (CNN) architecture as implemented in FedBN (see Table 4). This choice was made to recreate the experiments conducted in FedBN for direct performance comparison.

Table 3: Model architecture used in the evaluation with the synthetic dataset.

| Layer | Details |
| --- | --- |
| 1 | Linear(input_dim, 32), ReLU |
| 2 | Linear(32, 16), ReLU |
| 3 | Linear(16, 1), Sigmoid |
| 4 | Output Layer |

Table 4: Model architecture used in the evaluation with Office-31 dataset, as implemented in FedBN (Li et al., 2021b)

| Layer | Details |
| --- | --- |
| 1 | Conv2D(3, 64, kernel_size=11, stride=4, padding=2), BatchNorm(64), ReLU, MaxPool2D(kernel_size=3, stride=2) |
| 2 | Conv2D(64, 192, kernel_size=5, padding=2), BatchNorm(192), ReLU, MaxPool2D(kernel_size=3, stride=2) |
| 3 | Conv2D(192, 384, kernel_size=3, padding=1), BatchNorm(384), ReLU |
| 4 | Conv2D(384, 256, kernel_size=3, padding=1), BatchNorm(256), ReLU |
| 5 | Conv2D(256, 256, kernel_size=3, padding=1), BatchNorm(256), ReLU, MaxPool2D(kernel_size=3, stride=2) |
| 6 | AdaptiveAvgPool2D((6, 6)) |
| 7 | Linear(256 * 6 * 6, 4096), BatchNorm1d(4096), ReLU |
| 8 | Linear(4096, 4096), BatchNorm1d(4096), ReLU |
| 9 | Linear(4096, num_classes) |
| 10 | Output Layer |

## B.3 Results in different non-IID scenarios

Table 5 presents the results for the feature distribution skew scenario, where clients have heterogeneous feature distributions. FedMAP consistently improves upon individual client training for all clients, validating its effectiveness in handling diverse feature distributions through personalized client models while leveraging the guidance of the global prior.

Table 6 shows the results for the quantity skew scenario, where some clients (6-10) have limited sample sizes (highlighted in red). FedMAP provides significant gains of up to 8.90% over individual

Table 5: Validation accuracies of FedMAP, FedBN, FedProx, FedAvg, and individual training on the synthetic dataset with feature distribution skew. Each client has a unique transformation applied to its local data, introducing heterogeneity in the feature space.

| Client | Individual | FedMAP | FedBN | FedProx | FedAvg |
|--------|-----------|--------|-------|---------|--------|
| 1 | 82.25% | 85.26% (↑ 3.01%) | 62.77% | 58.44% | 57.85% |
|   | ±0.44% | ±0.10% | ±0.39% | ±0.07% | ±0.15% |
| 2 | 86.24% | 88.94% (↑ 2.70%) | 65.86% | 64.28% | 65.35% |
|   | ±0.31% | ±0.09% | ±0.34% | ±0.07% | ±0.15% |
| 3 | 80.76% | 83.70% (↑ 2.94%) | 66.99% | 65.99% | 66.06% |
|   | ±0.14% | ±0.13% | ±0.31% | ±0.14% | ±0.16% |
| 4 | 88.07% | 89.70% (↑ 1.63%) | 65.32% | 62.09% | 63.46% |
|   | ±0.15% | ±0.03% | ±0.31% | ±0.13% | ±0.14% |
| 5 | 83.03% | 84.97% (↑ 1.94%) | 66.46% | 64.78% | 64.74% |
|   | ±0.24% | ±0.12% | ±0.22% | ±0.07% | ±0.14% |
| 6 | 80.27% | 83.39% (↑ 3.12%) | 61.08% | 58.92% | 60.03% |
|   | ±0.24% | ±0.15% | ±0.25% | ±0.11% | ±0.09% |
| 7 | 83.48% | 86.55% (↑ 3.07%) | 68.39% | 67.34% | 66.65% |
|   | ±0.21% | ±0.05% | ±0.52% | ±0.17% | ±0.10% |
| 8 | 82.21% | 85.38% (↑ 3.17%) | 64.47% | 61.25% | 61.54% |
|   | ±0.27% | ±0.11% | ±0.39% | ±0.07% | ±0.05% |
| 9 | 85.92% | 87.45% (↑ 1.53%) | 66.16% | 63.14% | 63.55% |
|   | ±0.16% | ±0.07% | ±0.16% | ±0.16% | ±0.24% |
| 10 | 85.78% | 87.38% (↑ 1.60%) | 66.33% | 63.77% | 62.53% |
|   | ±0.25% | ±0.14% | ±0.30% | ±0.05% | ±0.09% |
| **Average** | 83.38% | 86.27% | 65.38% | 63.00% | 63.10% |
|   | ±0.24% | ±0.10% | ±0.32% | ±0.10% | ±0.13% |

training for clients 6-10, demonstrating its ability to effectively leverage information from clients with larger sample sizes.

## B.4 ADDITIONAL EXPERIMENT

We performed additional experiments using the Federated Extended MNIST (FEMNIST) dataset provided by the LEAF framework (Caldas et al., 2019). FEMNIST consists of 62 different classes of handwritten characters (0-9, a-z, A-Z) collected from 3,500 writers, with a total of 805,263 samples. With LEAF's preprocessing tools, we partitioned the dataset into 36 subsets with non-IID settings. Each partition followed LEAF's default split ratio of 90% training and 10% validation data. We implemented a CNN architecture detailed in Table 7. The model was trained using the SGD optimizer with an initial learning rate of 0.001 and a batch size of 64. The experimental results in Table 8, which show averages of validation accuracies across all 36 clients, demonstrate that FedMAP outperformed other FL approaches and individual training. These results align with our findings from the synthetic and Office-31 experiments, further validating FedMAP's effectiveness in handling real-world FL scenarios with natural non-IID settings.

Table 6: Validation accuracies of FedMAP, FedBN, FedProx, FedAvg, and individual training on the synthetic dataset with quantity skew. Clients 1-5 have 2000 samples each, while clients 6-10 (red) have only 500 samples each.

| Client | Individual | FedMAP | FedBN | FedProx | FedAvg |
|---|---|---|---|---|---|
| 1 | 83.37% | 84.47% (↑ 1.10%) | 67.69% | 64.33% | 66.45% |
| | ±0.08% | ±0.13% | ±0.24% | ±0.08% | ±0.10% |
| 2 | 88.10% | 89.64% (↑ 1.54%) | 72.41% | 70.10% | 70.22% |
| | ±0.15% | ±0.08% | ±0.48% | ±0.10% | ±0.10% |
| 3 | 87.48% | 88.88% (↑ 1.40%) | 73.74% | 72.56% | 71.08% |
| | ±0.13% | ±0.06% | ±0.45% | ±0.06% | ±0.13% |
| 4 | 84.40% | 86.95% (↑ 2.55%) | 71.38% | 69.19% | 68.82% |
| | ±0.15% | ±0.16% | ±0.46% | ±0.09% | ±0.12% |
| 5 | 78.47% | 80.23% (↑ 1.76%) | 66.96% | 63.93% | 62.76% |
| | ±0.16% | ±0.07% | ±0.31% | ±0.17% | ±0.13% |
| 6 | 61.62% | 65.48% (↑ 3.86%) | 56.28% | 54.49% | 53.96% |
| | ±0.70% | ±0.34% | ±0.61% | ±0.11% | ±0.52% |
| 7 | 64.07% | 72.97% (↑ 8.90%) | 54.96% | 56.05% | 55.02% |
| | ±0.45% | ±0.15% | ±0.76% | ±0.14% | ±0.33% |
| 8 | 63.83% | 67.10% (↑ 3.27%) | 48.58% | 45.45% | 46.33% |
| | ±0.31% | ±0.15% | ±0.53% | ±0.20% | ±0.32% |
| 9 | 67.32% | 74.92% (↑ 7.60%) | 62.26% | 60.37% | 61.03% |
| | ±0.37% | ±0.25% | ±0.82% | ±0.18% | ±0.17% |
| 10 | 64.04% | 71.74% (↑ 7.70%) | 61.99% | 57.03% | 59.73% |
| | ±0.49% | ±0.40% | ±0.69% | ±0.30% | ±0.14% |
| **Average** | 75.37% | 79.06% | 63.80% | 61.54% | 61.88% |
| | ±0.30% | ±0.18% | ±0.54% | ±0.14% | ±0.21% |

Table 7: Model architecture used in the experiment with FEMNIST dataset.

| Layer | Details |
|---|---|
| 1 | Conv2D(1, 32, kernel_size=3, padding=1), BatchNorm2d(32), ReLU |
| 2 | Conv2D(32, 64, kernel_size=3, padding=1), BatchNorm2d(64), ReLU, MaxPool2D(kernel_size=2, stride=2) |
| 3 | Conv2D(64, 64, kernel_size=3, padding=1), BatchNorm2d(64), ReLU |
| 4 | Conv2D(64, 128, kernel_size=3, padding=1), BatchNorm2d(128), ReLU, MaxPool2D(kernel_size=2, stride=2) |
| 5 | Flatten Layer |
| 6 | Linear(128 * 7 * 7, 512), BatchNorm1d(512), ReLU, Dropout(p=0.5) |
| 7 | Linear(512, 62) |
| 8 | Output Layer |

Table 8: Validation accuracies of individual training, FedMAP, FedBN, FedProx and FedAvg and on FEMNIST

| Approach | Individual | FedMAP | FedBN | FedProx | FedAvg |
|---|---|---|---|---|---|
| **Avg Accuracy** | 78.64% | 84.15%(↑ 5.51%) | 78.37% | 79.37% | 78.06% |

