# OpenReview forum: "FedMAP: Unlocking Potential in Personalized Federated Learning through Bi-Level MAP Optimization"
_ICLR.cc/2025/Conference — Submitted to ICLR 2025_

### Official Review · Reviewer_1LZX · 2024-10-29

**Soundness:** 2
**Presentation:** 2
**Contribution:** 3
**Rating:** 6
**Confidence:** 2

**Summary:**

This work introduces FedMAP, a Bayesian Personalized Federated Learning (PFL) framework that addresses data heterogeneity in federated learning (FL) by using Maximum A Posteriori (MAP) estimation. FedMAP applies a parametric prior distribution updated during aggregation, allowing it to handle non-IID data across clients.

**Strengths:**

The paper takes an alternative approach to addressing the non-iid data in federated learning though a Bayesian approach, which might be of the interest to the community.

**Weaknesses:**

The paper in its current form suffers from the following weaknesses:

1) The flow of the technical contents is extremely hard to follow. To provide some context, I actively publish in the federated learning domain and I could still not follow the contents. To be more precise, it will only take 10-11 lines after the beginning of Section 2 until the reader gets lost. This is because an alternative representation of federated learning process is studied, which is unconventional to the people in this area, without building the background for the reader. To address this, I suggest that authors provide more background on their alternative representation of federated learning early in Section 2, and include a high-level overview of their approach before diving into technical details.

2) The theoretical results of the paper are quite obscure. In particular, they are all provided in the appendix. Even the statement of the main theorems are not mentioned in the main text, which makes following the text even harder. To address this, I suggest that the authors bring some of the theoretical results (at least the main ones) to the main text and add explanations about them.

3) The simulation results are not well justified. After reading the sections I was left with several major questions:

3-1) Why standard federated learning datasets (MNIST, Fashion MNIST, Federated MNIST, SVHN, CIFAR-10, CIFAR-100) are not considered? Please add justifications about this.

3-2) Why standard personalized federated learning frameworks (mentioned in line 94-105) are not considered for performance comparison and only the most naive methods of FL (that by the way are not personalized FL), i.e., FedAVG and FedProx, are considered in conjunction with FedBN? In particular, the pressing need for having meta-learning based approaches as baselines is not addressed. Please add justifications about this issue to the paper.

**Questions:**

Please refer to my comments above about the weaknesses of the paper.

---

> ### Author Response · Authors · 2024-11-20
>
> We thank the reviewer for their careful consideration of our paper and detailed comments. We address each of their concerns below.
>
> 1. On reflection, we agree that Section 2, in particular, could be more accessible. In the revised version, we will update the section to provide a high-level overview and build intuition before introducing the technical details.
>
> 2. We agree that theoretical analysis should be included in the main paper. In the revised manuscript, we will integrate the Theorem 1 and its implications directly into the main text.
>
> 3.1. We agree that evaluating on more standard benchmarks is valuable. We chose Office-31 as it is commonly used in approaches that address feature distributions and naturally introduces real-world data heterogeneity through its three distinct domains, which includes realistic distribution shifts in feature distributions, quantities, and label distributions. Moreover, we used synthetic datasets that allow controlled analysis of specific non-IID scenarios to confirm FedMAP's effectiveness. We are currently performing the experiments on larger datasets such as FEMNIST and GLUE.
>
> 3.2. Our initial experiments focused on methods addressing similar challenges and were integrated within the Flower framework to ensure fair comparison and reproducibility. We agree that comparing against additional personalized FL approaches would strengthen our evaluation, and we are extending our comparisons to include meta-learning based approach FedL2P[1] in the revised version.
>
> Reference:
>
> [1] Lee, R., Kim, M., Li, D., Qiu, X., Hospedales, T., Huszár, F., \& Lane, N. D. (2023). FedL2P: Federated Learning to Personalize. Proceedings of the Thirty-seventh Conference on Neural Information Processing Systems.

---

> > ### Comment · Reviewer_1LZX · 2024-11-21
> >
> > Thanks for the response! If the authors edit the paper as promised (and upload a revised version of the paper) and conduct the experiments, I would be willing to change my score.

---

> > > ### Author Response · Authors · 2024-11-29
> > >
> > > We thank the reviewer once again for their valuable feedback and thoughtful suggestions, which have significantly contributed to improving our manuscript. We have addressed your concerns in this revised version and share the updates below for completeness.
> > >
> > > 1. We have simplified the presentation of our approach in section 2, skipping unnecessary technical details and giving more intuition. In the new version, we clearly present the updated rule for the local training and aggregation in equations (2) and (3) respectively, and provide some intuition behind these equations. Section 2.3 is devoted to the main theoretical contribution, which proves that the FedMAP iterations correspond to gradient descent iterations applied to the strongly convex functional given in equation (4). We hope the reader finds it more accessible and engaging than in the former version.
> > >
> > > 2. We have integrated the statement of Theorem 1 in the main paper (section 2.3), along with a short discussion before and after the statement, explaining the main implications of this result. Further details and the proof of the Theorem can still be found in Appendix A.
> > >
> > > 3. As promised, we ran additional experiments to address concerns regarding the evaluation scope. Using the FEMNIST dataset, which includes 805,263 samples from 3,500 writers cross 62 classes with a non-IID split, FedMAP achieved an accuracy of 84.15\%. This outperforms traditional FL approaches, including FedAvg (78.06\%), FedProx (79.37\%), FedBN (78.37\%), as well as a meta-learning-based PFL FedL2P (81.4\%). Because we were limited by time, we were not able to test on other standard datasets. However, we believe these results, along with our synthetic and Office-31 datasets evaluation results, provide complete validation of FedMAP's effectiveness in handling real-world non-IID scenarios across different scales.
> > >
> > > We hope that now we have addressed your concerns and performed additional experiments demonstrating the merits of our approach that you may consider revising your score.

---

> > > > ### Comment · Reviewer_1LZX · 2024-12-02
> > > >
> > > > Thanks for the effort in addressing my comments. The writing of the paper has got better but the text is still hard to follow. Given the improvements, I increase my score slightly but I think major changes might be needed to this paper before it becomes readable for a general audience working on federated learning.

---

### Official Review · Reviewer_LLiC · 2024-11-04

**Soundness:** 1
**Presentation:** 2
**Contribution:** 1
**Rating:** 3
**Confidence:** 5

**Summary:**

The authors propose FedMAP, a Bayesian PFL framework which applies Maximum A Posteriori (MAP) estimation to effectively mitigate various non-IID data issues, by means of a parametric prior distribution, which is updated during aggregation. In this FL approach, the authors formulate the local training problem as a MAP estimation of the local models, in which the global model acts as a prior distribution on the hypothesis set of probabilistic models.

**Strengths:**

- The framework applies Maximum A Posteriori (MAP) estimation to tackle non-IID problems in the personalized FL framework.

- The problem formulation is clearly described.

**Weaknesses:**

- The idea of personalized FL as bi-level optimization with Moreau envelops was proposed first in the pFedMe paper (https://proceedings.neurips.cc/paper/2020/hash/f4f1f13c8289ac1b1ee0ff176b56fc60-Abstract.html).  Basically, this work decorated pFedMe with MAP, especially the Local Optimization at eq. (7). Therefore, the contribution of this work is very limited. But ironically, this paper did not cite pFedMe for some reason.

- The theoretical results did not show why the global aggregation (line 5, Algorithm 3) can converge to the unique optimal solution of the bilevel optimization problem with Moreau envelop.

- In the experiments, the work compared the framework only with a very old framework like FedAvg and two other methods like Fedbn and FedProx. However, the work did not compare the result with similar approaches (pFedMe, Ditto, FedDyn).

- The datasets used for experiments are relatively small (only 4110 images). Similar FL experiment setups used much larger datasets (pFedMe, Ditto, FedDyn), eventually regarding non-IID problems. We suggest that the authors test with popular and large FL datasets, like GLUE and FEMNIST.

**Questions:**

- Can you provide a detailed comparison between FedMAP and pFedMe by highlighting any key differences or improvements? Why was pFedMe not cited, and how does this work build upon or differ from this prior research?

- The paper uses only small datasets to test the benchmark of the proposed approach. How is the non-IID problem improved with a relatively small dataset?

- Why did you not compare the benchmark with pFedMe/Ditto/FedDyn while using the same technique at local rounds?

- Do you have any theoretical results to show that the global aggregation (line 5, Algorithm 3) can help the algorithm converge to the unique optimal solution of the bilevel optimization problem with the Moreau envelop?

- If not, can you provide additional theoretical analysis specifically addressing the convergence properties of the global aggregation step?

- Can you discuss the scalability of FedMAP to larger datasets? For example, can you conduct experiments on GLUE and FEMNIST or provide a detailed justification for why the current datasets are sufficient to demonstrate the effectiveness of FedMAP, particularly in non-IID settings?

---

> ### Author Response · Authors · 2024-11-16
>
> We thank the reviewer for their time and detailed review. They raised a critical oversight in our omission of pFedMe, we were unaware of this paper and acknowledge that at an initial look, it seems to be very similar to our work and should have been referenced. Below, we highlight in detail how pFedMe is a special case of FedMAP, rather than being equivalent, and that FedMAP is a significantly more general framework which we would exploit in a revised version.
>
> Q1: The oversight in citing pFedMe was a good-faith error, we completely agree that this should have been cited as it is an important contribution, also being a special case of FedMAP, it is entirely relevant for citation and will be corrected in the revised manuscript.
> While we acknowledge the mathematical similarity between pFedMe and FedMAP, we stress that this similarity only arises for a specific case of FedMAP.
> Indeed, the regularizer $\mathcal{R} (\cdot, \gamma)$ in FedMAP is a parametrized function with parameter $\gamma$.
> When $\mathcal{R} (\cdot, \gamma)$ is taken as the negative log-likelihood of a Gaussian distribution, where the parameter $\gamma$ is the mean, the regularizer is of the form $\mathcal{R} (\theta, \gamma) = \alpha \|\theta - \gamma\|^2$, for some $\alpha>0$ fixed. In this case, FedMAP coincides with pFedMe.
> We admit that in the present version of the paper, we mainly considered this choice, and therefore it is not clear what are the differences between pFedMe and FedMAP.
> However, one can consider other parametrizations for the regularizer, which offer significant advantages over the quadratic case with a fixed hyperparameter $\alpha$.
>
> For instance, one of the key considerations when using a quadratic regularizer is the choice of $\alpha$, which is related to the variance of the parameters of the model across the clients.
> When the model, $\Phi( \cdot; \theta)$ has $p$ parameters, i.e. $\theta\in \mathbb{R}^p$, one can consider the FedMAP approach with a regularizer of the form
> \begin{equation}
> \mathcal{R} (\theta, (\gamma_1, \gamma_2)) = \sum_{i=1}^p \alpha (\gamma_2^{(i)} ) \left( \theta_i - \gamma_1^{(i)} \right)^2. \tag{1}
> \end{equation}
> where $\alpha: \mathbb{R} \to \mathbb{R}^+$ is a function that has to be suitably chosen in a way that the function $(\gamma_1, \gamma_2) \mapsto \mathcal{R} (\theta, (\gamma_1, \gamma_2))$ is convex for all $\theta$.
> With this choice, the parameter $\alpha$, associated with the variance of each model parameter, is updated during the aggregation step according to the local models obtained from the different clients.
> This is important since some parameters might be similar across the local models whereas others might be needed to be different. This represents a significant improvement over the pFedMe approach.
>
> In a revised version of the paper, we will now consider the more general regularizer, as shown in Eq.~(1), instead of the quadratic case for the parameterized regularizer, and describe precisely how to choose the function $\alpha(\cdot)$.
>
> Q2: The current experiments show the effectiveness of FedMAP in handling non-IID challenges in two ways. We used synthetic datasets that were specifically constructed to show the controlled combinations of non-IID issues (feature, label, and quantity skew). We also used Office-31 datasets which naturally introduced the feature distribution skewness across three different domains despite their size being small. However, as recommended by the reviewer, we will evaluate FedMAP with large datasets FEMNIST and GLUE to strengthen our findings.
>
> Q3:In this paper, we focused on the foundational method such as FedAvg to establish baseline performance. FedProx was chosen because it particularly addresses client drift with regularization. We included FedBN because it specifically addresses feature shift challenges, which are key issues FedMAP aims to address.
> However, on reflection, we agree with the reviewer that comparing with pFedMe, Ditto, and FedDyn would provide a more comprehensive evaluation.
> We have already started these experiments and in the revised manuscript, we will add these additional baselines to our experiments and report the results.
>
> Q4: In the current version, Algorithm 3 represents the aggregation step for the specific case when $\mathcal{R} (\theta, \gamma) = \| \theta - \gamma\|^2$. In a more general setting, this should be replaced by a gradient step as in equation (11) in the paper. Convergence of the FedMAP iterations is guaranteed since, as shown in Theorem 1, these iterations correspond to gradient descent applied to a convex function. We will make sure this is clear to the reader in the revised version of the manuscript.
>
> Q5: This has been addressed in the response to Q4 above.
>
> Q6: While Office-31 is common for demonstrating feature shifts, we agree to include larger datasets (FEMNIST and GLUE) in our revised experiments.

---

> > ### Comment · Reviewer_LLiC · 2024-11-23
> > **Feedback**
> >
> > Thanks for the response. Even though there’s oversight of missing pFedMe citation and thus limited novelty, I’m happy to increase the score.

---

> > > ### Author Response · Authors · 2024-11-29
> > >
> > > We thank the reviewer again for their time and efforts in improving our manuscript. We have now updated the manuscript and addressed their comments as follows:
> > >
> > > Q1: We have corrected our error in the omission of the pFedMe paper [T Dihn et al. (2020)]. A reference to this paper can be found in the related literature section 1.1 (see line 126). We also talk about pFedMe in section 2.3, as a special case of our approach.
> > >
> > > A more detailed discussion about how FedMAP is significantly more general and flexible approach to PFL than pFedMe can be found in Appendix A.2, which narrowly considers the case of quadratic regularizers.
> > >
> > > We have improved this section by considering a more flexible regularizer parametrized not only by the mean, but also the variance of the associated Gaussian distribution (see the paragraph starting in line 814 and equation (18). The construction of such a parameterization is provided in the new Lemma 1, where we prove that the convexity conditions of Theorem 1 are fulfilled.
> > >
> > > We have also updated the linear regression example in Appendix A.4. We now consider the parametric regularizer presented in A.2, in which we parametrized the mean and the variance of the associated Gaussian prior. We show that, during the FedMAP iterations, the algorithm is able to learn which parameters in the linear regression model must have a larger variance.
> > >
> > > Q2: We have run additional experiments on FEMNIST, which contains 805,263 samples from 3,500 writers across 62 classes. Our FEMNIST experiments show FedMAP achieves 84.15\% accuracy, significantly outperforming both individual training (78.64\%) and other approaches (FedBN: 78.37\%, FedProx: 79.37\%, FedAvg: 78.06\%). We also compared against a meta-learning based PFL approach called FedL2P which achieved 81.4\% accuracy. The improvement over individual training demonstrates FedMAP's ability to leverage collaborative learning while maintaining personalization. Due to time constraints for the submission, we were unable to complete evaluations on GLUE dataset. However, we believe the combination of FEMNIST results and synthetic experiments can confirm FedMAP's effectiveness in handling non-IID data at different scales.
> > >
> > > Q3: Although pFedMe is a single special case of the FedMAP framework, their implementations are fundamentally different. Due to reproducibility challenges with the pFedMe codebase on the FEMNIST dataset, we were unable to conduct direct performance comparisons. However, we identified several key differences between the approaches. FedMAP implements personalization through a regularization term in the loss function, whereas pFedMe does this with optimizer customization which incorporates a proximal term during the optimization process. Also, pFedMe resets local models to the global state after aggregation, potentially reducing local adaptations. In contrast, FedMAP keeps and builds on previously trained personalized models, enabling continuous refinement of client-specific features across communication rounds.
> > >
> > > Q4: The main theoretical contribution (Theorem 1) has been moved to the main paper, section 2.3. We have modified the entire section 2. In the new version, we clearly explain how the  FedMAP iterations, given by equations (2) and (3), are guaranteed to converge since they correspond to gradient descent iterations of a strongly convex functional, given in equation (4). A more detailed discussion, along with the proof of the main Theorem, is provided in Appendix A.
> > >
> > > We hope that with these additional experiments and revisions to the paper demonstrating the novelty of the approach and the significant gains resulting from the flexibility we introduce that you may consider revising your score.

---

### Official Review · Reviewer_aR5d · 2024-11-04

**Soundness:** 3
**Presentation:** 4
**Contribution:** 3
**Rating:** 6
**Confidence:** 3

**Summary:**

The paper presents FedMAP, a framework designed for Personalized Federated Learning (PFL) to handle the challenges of non-IID (non-identically distributed) data across clients. FedMAP integrates Maximum A Posteriori (MAP) estimation into the federated learning process, allowing each client to personalize its model by updating a shared global prior with its local data through a bi-level optimization procedure.

The approach aims to improve on standard federated learning methods, which often struggle with heterogeneous data, by balancing global knowledge sharing with local adaptation. The paper provides a theoretical analysis of the convergence properties of FedMAP and evaluates its performance through experiments on non-IID datasets, comparing it to existing federated learning methods, such as FedAvg, FedProx, and FedBN.

**Strengths:**

- The writing is clear and the presentation is well-structured, making it easy for readers to follow and understand the proposed approach and its contributions.
- Bi-Level Optimization Framework: The paper provides a theoretical analysis based on bi-level optimization, offering insights into how the proposed FedMAP converges under heterogeneous data conditions.
- Although not ground-breaking, the convergence analysis (in Appendix A) is nice. I think it is better to place somewhere in the main text rather than in the appendix.
- FedMAP can be applied in various federated learning scenarios with minimal changes to the standard federated learning process, making it adaptable for different use cases.
- The experimental results show that FedMAP outperforms traditional federated learning methods in a range of non-IID scenarios, demonstrating improved performance in tasks involving skewed or imbalanced data distributions.

**Weaknesses:**

- The proposed method assumes an isotropic Gaussian prior for the global model, and the paper does not explore how the choice of prior might affect the model's performance or consider alternatives that could be more suited for specific tasks. Can the results be relaxed to the setting where the prior is still Gaussian  with diagonal variance-covariance, but non-isotropic Gaussian?
- The paper does not explicitly mention whether FedMAP reduces communication costs compared to existing methods. Instead, the focus is on improving model performance in non-IID settings through bi-level optimization and personalization. While these methods might introduce computational overhead, the communication efficiency aspect is not fully addressed. Can you elaborate on this?

**Questions:**

See the questions in the weakness section.

---

> ### Author Response · Authors · 2024-11-20
>
> We would like to thank the reviewer for their thoughtful comments. Below are our detailed responses to each point raised.
>
> Q1: We would like to clarify the choice of the prior distribution. In Section 3, we present an implementation of FedMAP using a normal distribution as the prior, in which the parameter is the mean. This choice is made for practical demonstration because of its simplicity. However, FedMAP is designed as a more general framework. In Section 2, we present our method for any parameterized family of probability measures over the parameter space. The theoretical foundations in Section 2 and Theorem 1 are established for this general setting, with the requirements being that the convexity condition in Assumption 1 holds. The Gaussian implementation was chosen only for its simplicity and interpretability, but the framework can accommodate non-isotropic priors, non-Gaussian parametric families, and task-specific prior distributions that better capture domain knowledge.
> In the revised version, we will consider, as an example, a more flexible parameterization of a Gaussian prior. Namely, we shall consider a Normal distribution, in which the parameters will be both the mean and the variance. This choice lifts the requirement of having to manually choose the variance of the prior as a hyperparameter.
>
> Q2: FedMAP has the same communication complexity as FedAvg. In each round, the server broadcasts the parameters $\gamma$ defining the current prior distribution, and each client sends back their local parameters $\theta_k$ and weight $\omega_k$. FedMAP indeed introduces additional computational overhead in the local optimization step through MAP estimation. However, this computational cost is offset by improved communication efficiency. As shown in Theorem 1, the FedMAP updates correspond to gradient descent iterations for a strongly convex function $M(\gamma)$, which ensures faster convergence with fewer communication rounds. In non-IID settings, client drift often requires more rounds using traditional methods. The trade-off between increased local computation and reduced communication rounds is especially beneficial in FL where communication is often the primary bottleneck. We agree that analyzing the trade-offs is important for practical deployment and plan to include detailed empirical analysis in the revised version of the manuscript.

---

> > ### Comment · Reviewer_aR5d · 2024-11-25
> >
> > I appreciate the responses. They have mostly addressed my questions. I will maintain my score for now as it already leans towards acceptance.

---

> > > ### Author Response · Authors · 2024-11-29
> > >
> > > Once again, we appreciate your review and positive assessment of our manuscript along with your response to our comments. We are grateful for your recognition of our paper's clear presentation and theoretical foundations. We have addressed your specific concerns in our revised version of the manuscript and just wanted to highlight these to you for completeness:
> > >
> > > Q1: We have expanded our framework beyond the isotropic Gaussian implementation. In Section 2.2, we still consider the case of general priors with learnable parameters. We updated Section 2.3 with Theorem 1 which shows theoretical guarantees with more generic parametric families. In the first version, this Theorem was only included in the appendix.
> > >
> > > All the details about the specific case of Gaussian priors (or quadratic regularizers) are included in Appendix A.2. This section has been improved by considering general (non-isotropic) Gaussian priors with learnable mean and variance (see line 814 and equation(18)). In the new Lemma 1, we provide the construction of such parametrization and prove that it satisfies the convexity conditions of Theorem 1. We have also updated the linear regression example in appendix A.4, in which we now consider the parametric Gaussian prior from A.2. We show that, during the FedMAP iterations, the algorithm is able to learn which parameters in the linear regression model must have a larger variance.
> > >
> > > Q2: Although the MAP estimation introduces some computational overhead locally, our theoretical results in Theorem 1 show that FedMAP's gradient descent iterations on a strong convex function have faster convergence with fewer communication rounds.
> > >
> > > We hope that with these revisions you may consider revising your score.

---

### Official Review · Reviewer_nmcf · 2024-11-07

**Soundness:** 3
**Presentation:** 3
**Contribution:** 2
**Rating:** 5
**Confidence:** 3

**Summary:**

The paper proposes a new personalized federated learning algorithm based on maximum a posteriori (MAP). Each client constructs the personalized model given global model as prior.

**Strengths:**

The paper proposes a novel federated learning algorithm. In appendix the paper analyses the convergence of the algorithm. The performance of the proposed algorithm is compared against some well-known federated learning algorithms on common datasets.

**Weaknesses:**

- The paper should provide more concrete and technical explanation about their novelty compared to existing works. Reading the related works section, I believe the discussion is incomplete and the advantages of FedMAP compared to other methods are not clear. I suggest to provide such discussion after introducing the proposed algorithm.
- Although the paper provides some theoretical analyses, these analyses are in appendix and not in the main text of the paper. This makes these analyses disconnected from the paper for readers. It is not clear how important the theoretical contributions of this paper is. I suggest makes these conclusions from the theoretical analysis more clear. This can better show the contribution of the paper relative to other works.
- I felt that the literature review of the paper is not very up-to-date. This can be improved by including more 2024 papers in the related works section.
- The experiments can benefit from adding more baselines although I think it may not be necessary.

**Questions:**

Can you please give more explanation about the challenges and problems that is solved by FedMAP which cannot be solved by other personalized federated learning algorithms?

---

> ### Author Response · Authors · 2024-11-20
>
> We thank the reviewer for their thoughtful comments and efforts toward improving our manuscript. We are glad that you found our proposed algorithm novel and appreciated the suggested improvements. Below, we would like to address your main concerns:
>
> W1: We agree that our paper should provide a more technical discussion. We will add a new subsection that provides formal analysis comparing our bi-level optimization to the existing PFL approaches and demonstrate how FedMAP's envelope function and adaptive scheme facilitate better handling of non-IID data. Also, we added a brief discussion in our response to Q1, which shows FedMAP is different from other PFL approaches.
>
> W2: We will move the statements of the key results of the theoretical analysis from the appendix to the main paper. The new section will present the convergence guarantees and connect theoretical results to practical benefits in non-IID settings.
>
> W3: We will update the related work section to include the relevant 2024 publications such as PerAda[1], FedASA[2] and pFedEM[3].
>
> W4: Although our current experimental evaluation includes both conventional and personalized FL baselines, and their comparison results present FedMAP outperforms, we agree that additional baselines will strengthen our argument. We will include more recent and relevant PFL approaches in our evaluations.
>
> Q1: Other PFL approaches mentioned in our related work section include fine-tuning based approaches, which use sequential optimization (first global, then local) without regularization, whereas FedMAP provides continuous adaptation with its learned prior. Layer-wise based methods have constraints on specific layers/components, but FedMAP allows continuous, full-model adaption. Compared to meta-learning based approaches which require complex second-order optimization, FedMAP is more efficient with first-order updates. In contrast to the clustering based which uses discrete clustering, FedMAP provides more smoother knowledge transfer using the adaptive weighing. The key novelty of FedMAP is its bi-level optimization framework. It minimizes
> \begin{equation}
> M(\gamma) = \sum_{k=1}^q M_k(\gamma; Z_k) = \sum_{k=1}^q \min_{\theta} \{L(\theta; Z_k) + R_k(\theta, \gamma)\}
> \end{equation}
> where the inner optimization personalizes local models and the outer optimization learns a global prior. This structure provides several advantages:
>
> 1. The framework offers convergence guarantees for any parameterized regularizer $R_k(\theta, \gamma)$ satisfying suitable convexity conditions (Theorem 1), making it general and flexible for different personalization strategies.
>
> 2. The local optimization
> \begin{equation}
> \theta_k^* = \arg\min_{\theta} L(\theta; Z_k) + R_k(\theta, \gamma)
> \end{equation}
>
> enables client-specific adaptation while maintaining theoretical properties through MAP estimation.
>
> 3. The global parameter optimization through $M(\gamma)$ balances local personalization and global knowledge sharing, which is important for heterogeneous data distributions as demonstrated in our example for a linear regression task.
>
> References:
>
> [1] Xie, C., Huang, D.-A., Chu, W., Xu, D., Xiao, C., Li, B., \& Anandkumar, A. (2024). PerAda: Parameter-efficient federated learning personalization with generalization guarantees. *Proceedings of the IEEE/CVF Conference on Computer Vision and Pattern Recognition (CVPR)*, 23838–23848.
>
> [2] Deng, D., Wu, X., Zhang, T., Tang, X., Du, H., Kang, J., Liu, J., \& Niyato, D. (2024). FedASA: A personalized federated learning with adaptive model aggregation for heterogeneous mobile edge computing. *IEEE Transactions on Mobile Computing, 23*(12), 14787–14802. https://doi.org/10.1109/TMC.2024.3446271
>
> [3] Chen, S., Liu, W., Zhang, X., Xu, H., Lin, W., \& Chen, X. (2024). Adaptive personalized federated learning for non-IID data with continual distribution shift. *2024 IEEE/ACM 32nd International Symposium on Quality of Service (IWQoS)*, 1–6. https://doi.org/10.1109/IWQoS61813.2024.10682851

---

> > ### Author Response · Authors · 2024-11-29
> >
> > Again, we thank the reviewer for their thoughtful comments and efforts in improving our manuscript. Following our prior response, we have now also addressed your concerns in our revised version uploaded here:
> >
> > W1: We have modified Section 2 of the main paper, in which we now present the main theoretical contribution (Theorem 1). In the first version, this theorem was only included in Appendix A.1. In the paragraph after the statement of Theorem 1, we highlight the main difference between FedMAP and most of personalized FL approaches (the details are referred to the appendix). Also, in the paragraph starting in line 224, we mention a similar PFL approach [T Dihn et al. (2020)], also based on the minimization of envelope functions. We demonstrate that our approach is much more general and that their approach is only a special case of FedMAP. However, due to space constraints, the detailed explanation is postponed to the appendix A.2.
> >
> > We have also modified Appendix A.2. The main improvement is that we now consider parametric Gaussian priors in which both the mean and the variance are parameterized (see line 814 and equation(18)). The choice of this parametric prior is provided in the new Lemma 1, where we prove that the convexity conditions of the main theorem are fulfilled. The example for linear regression in Appendix A.4 has also been updated by considering the quadratic regularizer from A.2 and Lemma 1.
> >
> > W2: We have moved key theoretical results from the appendix to section 2.3 on the main paper. In this section, we now present Theorem 1 detailing the convergence properties of FedMAP and providing clear connections between updates to the prior and gradient descent iterations.
> >
> > W3: We have updated the Related Literature section with relevant 2024 such as PerAda[1], FedASA[2] and pFedEM[3].
> >
> > Q1: Other PFL approaches are discussed in the Related Literature section 1.1. In section 2, we highlight the significant differences between our approach and other existing approaches (see the discussion before and after Theorem 1). A more detailed discussion is included in Appendix A. See A.1 for a general discussion, A.2 for the case of quadratic regularizers and A.4 for an illustrative example of a linear regression task.
> >
> > We hope that with these revisions you may consider revising your score.

---

### Meta-Review · Area_Chair_486h · 2024-12-20

**Metareview:**

(a) Scientific Claims and Findings:

- The paper proposes FedMAP, a bi-level Bayesian Personalized Federated Learning (PFL) framework using Maximum A Posteriori (MAP) estimation to address non-IID data challenges.
- Claims include improved performance over baseline FL methods (e.g., FedAvg, FedProx), theoretical convergence guarantees, and flexibility in handling heterogeneous data.
- Theoretical contributions (e.g., bi-level optimization and convergence analysis) are highlighted but remain poorly integrated into the main text.

(b) Strengths:
- Clear and structured presentation: The paper is well-organized, making it easier to understand the proposed approach and its contributions.
- Adaptable and general bi-Level optimization Framework: The theoretical analysis provides valuable insights into FedMAP's convergence under heterogeneous data conditions.
- Empirical performance: Experimental results show that FedMAP outperforms traditional federated learning methods in handling non-IID data, particularly for imbalanced or skewed distributions.

(c) Weaknesses:
- Novelty concerns: I agree with the reviewer concerns on novelty. While the framework proposed here is more general than some of the prior works, the authors only instantiate it with Gaussian priors negating all of the generality. While this was not raised by the reviewers, these other works also seem to have a high overlap with this work:
>  Ozkara, Kaan, et al. "A statistical framework for personalized federated learning and estimation: Theory, algorithms, and privacy." International Conference on Learning Representations (ICLR), 2023. Also some prior work by the same authors: https://arxiv.org/pdf/2207.01771

- Comparision: The authors do not compare against any of the other Bayseisian-based personalization methods. What about the generality of the current approach provides advantage?

- Lack of theoretical depth: The fundamental issue in personalization is how to automatically identify the structure present in the datasets i.e. in what way is the information common vs. unique in the clients. Do they share a common low dimensional representation, or perhaps similar classfiers after domain adaptation, or perhaps the client optima form a mixture of Gaussians, etc. The bayesian approach provideds no answers here - it requires the user to encode this knowledge using the gloabl prior. The generality of the framework proposed in of itself is an issue - how should I instantiate this framework? There is no theoretical or empirical guidance provided.

**Additional Comments On Reviewer Discussion:**

Points Raised by Reviewers:

- Citation and Novelty: Omission of key related works like pFedMe and limited novelty beyond existing frameworks.
- Baseline Comparisons and Datasets: Lack of comparisons with relevant Bayesian-based PFL methods and SOTA personalized federated learning algorithms. Also inadequate evaluation on large-scale, standard datasets (e.g., GLUE, CIFAR).
- Theoretical Integration: Key theoretical contributions were initially relegated to the appendix, limiting their accessibility and impact.

How the Authors Addressed These Points:
- Citation and Novelty: Acknowledged omission of pFedMe and added it as a special case of FedMAP.
- Added FEMNIST dataset to experiments, demonstrating superior accuracy (e.g., 84.15%). Promised further experiments on GLUE but could not complete them before the deadline. Discussed challenges in reproducing pFedMe results and outlined conceptual differences. Promised scalability analysis but results remained incomplete.
- Theoretical Integration: Moved Theorem 1 (main theoretical result) to the main text and improved the discussion. Expanded explanations of convergence properties and global aggregation guarantees in Section 2. Provided a high-level overview of FedMAP before diving into technical aspects.

Final Decision. While the authors made meaningful improvements during the rebuttal, the paper’s limited novelty, theoretical depth, and incomplete empirical evaluation prevent it from meeting the bar for acceptance.

---

### Decision · Program_Chairs · 2025-01-22

Reject